# CXCR3 enables recruitment and site-specific bystander activation of memory CD8+ T cells

Nicholas J. Maurice [1,2], M. Juliana McElrath[1,3,4,5,6], Erica Andersen-Nissen[1,7], Nicole Frahm [1] & Martin Prlic [1,2,4,8]*

Bystander activation of memory T cells occurs in the absence of cognate antigen during infections that elicit strong systemic inflammatory responses, which subsequently affect host immune responses. Here we report that memory T cell bystander activation is not limited to induction by systemic inflammation. We initially observe potential T cell bystander activation in a cohort of human vaccine recipients. Using a mouse model system, we then find that memory CD8+ T cells are specifically recruited to sites with activated antigen-presenting cells (APCs) in a CXCR3-dependent manner. In addition, CXCR3 is also necessary for T cell clustering around APCs and T cell bystander activation, which temporospatially overlaps with the subsequent antigen-specific T cell response. Our data thus suggest that bystander activation is part of the initial localized immune response, and is mediated by a site-specific recruitment process of memory T cells.

[1] Vaccine and Infectious Disease Division, Fred Hutchinson Cancer Research Center, Seattle, WA 98109, USA. [2] Molecular and Cellular Biology Program, University of Washington, Seattle, WA 98195, USA. [3] HIV Vaccine Trials Network, Fred Hutchinson Cancer Research Center, Seattle, WA 98109, USA. [4] Department of Global Health, University of Washington, Seattle, WA 98195, USA. [5] Department of Medicine, University of Washington, Seattle, WA 98195, USA. [6] Department of Laboratory Medicine, University of Washington, Seattle, WA 98195, USA. [7] Cape Town HIV Vaccine Trials Network Immunology Laboratory, Hutchinson Centre Research Institute of South Africa, 8001 Cape Town, South Africa. [8] Department of Immunology, University of Washington, Seattle, WA 98195, USA. *email: mprlic@fredhutch.org

The main purpose of memory T cells is to rapidly respond when antigen (Ag) is re-encountered, however memory T cells can also be activated in an inflammation-dependent, but Ag-independent manner[1–9]. This phenomenon is referred to as bystander activation and has been reported in the context of acute and chronic infections in the mouse model system[10] as well as in humans, including chronic hepatitis C virus (HCV) infection[11], acute and chronic human immunodeficiency virus (HIV) infection[12,13], acute Epstein–Barr virus infection[5], and acute hepatitis A virus infection[3]. Given that memory T cell bystander activation has been reported in context of these infections that are either systemic in nature or have systemic inflammatory effects, memory T cells seemingly become bystander-activated in a passive manner by responding to systemically available inflammatory cues.

Once bystander-activated, memory CD8$^+$ T cells acquire an effector T cell-like phenotype, including expression of granzyme B[4,5,7], and IFNγ[2,6,8]. The biological significance of acquiring cytotoxic effector function in the absence of cognate Ag was long unclear, until it was shown that bystander-activated memory CD8$^+$ T cells are capable of direct cytolysis of target cells[3,7,14]. Direct target cell killing depends on NKG2D (expressed on memory T cells)—NKG2D ligand (a family of stress-induced proteins) interactions, which was first demonstrated in a mouse model system[7] and more recently also shown using human T cells[3]. Importantly, this innate-like recognition of target cells does not appear to be nearly as efficient as T cell receptor (TCR)-mediated target cell killing[7], but the consequences for the host are still significant given the large number of memory T cells and their ability to produce other effector molecules such as IFNγ[2,7,10,15,16]. Bystander activation of T cells can be beneficial to the host as these cells contribute to early pathogen clearance[2,7,9]. However, bystander-activated T cells have also been shown to drive pathogenesis in the context of chronic infections[10,14,15]. Thus, bystander-activated T cells appear to be a double-edged sword with benefits for the host when activation is brief (early pathogen control during acute infection) and detrimental when activation is persistent (tissue damage during chronic infections). This includes IL-15-driven differentiation of T cells into an NK-like cell type during Celiac disease[17].

There is a key role for IL-12, IL-15, and IL-18 in activating memory T cells and a central role for IFNγ in orchestrating the subsequent immune response[8]. Given the systemic nature of most infections studied so far, the existence of signals that could potentially recruit memory T cells to more localized sites of inflammation have not been investigated. Interestingly, recent data suggested that bystander activation may also occur when inflammation is localized, such as the tumor microenvironment with a T cell infiltrate consisting of tumor-specific and non-specific T cells[18]. The mechanisms that would allow bystander activation of memory T cells in such a scenario and the biological consequences of bystander activation are unclear.

Bystander activation of memory CD8$^+$ T cells occurs in the presence of systemic inflammation, presumably because memory T cells are exposed to available inflammatory cytokines. Using samples from a human vaccine trial we find evidence that suggests bystander activation of memory CD8$^+$ T cells may also occur shortly after vaccination. We use a mouse model to follow-up on this observation in an effort to determine whether and how bystander activation of memory CD8$^+$ T cells can occur in the context of spatially localized inflammation. We demonstrate that bystander activation during localized inflammation hinges on the ability of memory CD8$^+$ T cells to rapidly migrate to sites of early immune activation in a CXCR3-dependent manner. Thus, our data suggest that memory T cell bystander activation is not a passive event limited to scenarios where inflammatory cues are widely available as shown so far, but an active, migration-driven process. Importantly, at least some bystander-activated T cells remain located at these immune activation sites for days and temporally and spatially overlap with incoming Ag-specific T cells. We discuss the relevance of these bystander-activated T cells as the result of a localized and site-specific recruitment process for memory T cells for the subsequent host immune response.

## Results

**Evidence of bystander activation following vaccination**. We initially asked if we could detect any evidence of bystander activation of memory CD8$^+$ T cells in a human cohort following immunization with a live-attenuated vaccine. We analyzed the activation profile of human immune cells isolated from peripheral blood mononuclear cells (PBMCs) immediately prior to and 3 days after intra-muscular (i.m.) immunization with a modified vaccinia Ankara (MVA) vector using samples of the HIV Vaccine Trial Network (HVTN) 908 clinical trial (Supplementary Fig. 1a, b)[19,20]. We found that memory (CD45RO$^+$) CD8$^+$ T cells displayed significantly increased median fluorescence intensities (MedFI) for granzyme B in MVA/HIV62 but not placebo recipients (Supplementary Fig. 1c–e). Importantly, memory CD8$^+$ T cells (bulk or granzyme B$^+$) did not show increased expression of markers indicative of recent TCR signaling, such as 4-1BB[21] or PD-1[22] (Supplementary Fig. 1f–h). Similarly, mucosal-associated invariant T cells can also be bystander-activated[23–25] and showed a significant increase in CD69 expression, indicating that an i.m. administered vaccine may be sufficient to activate cells in an inflammation-dependent manner (Supplementary Fig. 1i). We next used a mouse model system to determine if bystander activation can occur as a result of localized inflammation.

**Ag-nonspecific T cells cluster at sites of early immune activation**. We wanted to visualize where and how early bystander activation occurs in spatial relation to activated APCs and Ag during the nascent stages of a localized immune response. To develop a traceable population of memory CD8$^+$ T cells with defined Ag-specificity, we adoptively transferred naive, congenically marked OT-I CD8$^+$ T cells, which express a TCR that recognizes the SIINFEKL peptide of chicken egg ovalbumin (OVA), into wild-type (WT) C57BL/6J recipients (Fig. 1a). We subsequently infected recipient mice with OVA-expressing vesicular stomatitis virus (VSV-OVA). This acute infection provides cognate Ag and inflammatory cues to generate effector OT-I T cells, which then contracted into a stable pool of memory OT-I T cells 60 days post infection (referred to as OT-I memory mice)[26]. As our main goal was to interrogate how memory T cells become activated in a localized manner, we tested several low-dose immunization strategies with actA$^-$ and wild-type (WT) Listeria monocytogenes (LM). OT-I T cells are not activated by any Listeria-derived Ags[7], thus any augmentation of OT-I T cell function would be due to inflammation-driven bystander effects (Fig. 1b, Supplementary Fig. 2a). We immunized OT-I memory mice with 10$^6$ colony forming units (cfu) actA$^-$ LM to mimic the live-attenuated human MVA vaccine. However, within 24 h of immunization with actA$^-$ LM nearly all white pulps (WP) in the spleen stained positive for LLO and were enriched for memory OT-I T cells (Supplementary Fig. 2b). A further titration in dose did not alleviate this problem, so this approach was not suitable to examine site-specific bystander activation of memory T cells. We next immunized mice with 1000 cfu of WT LM. This low challenge dose initially resulted in a very localized infection as infected APCs migrate to the periarteriolar lymphoid sheath inside the splenic WP within 6–12 hours[27–29] and was thus ideal

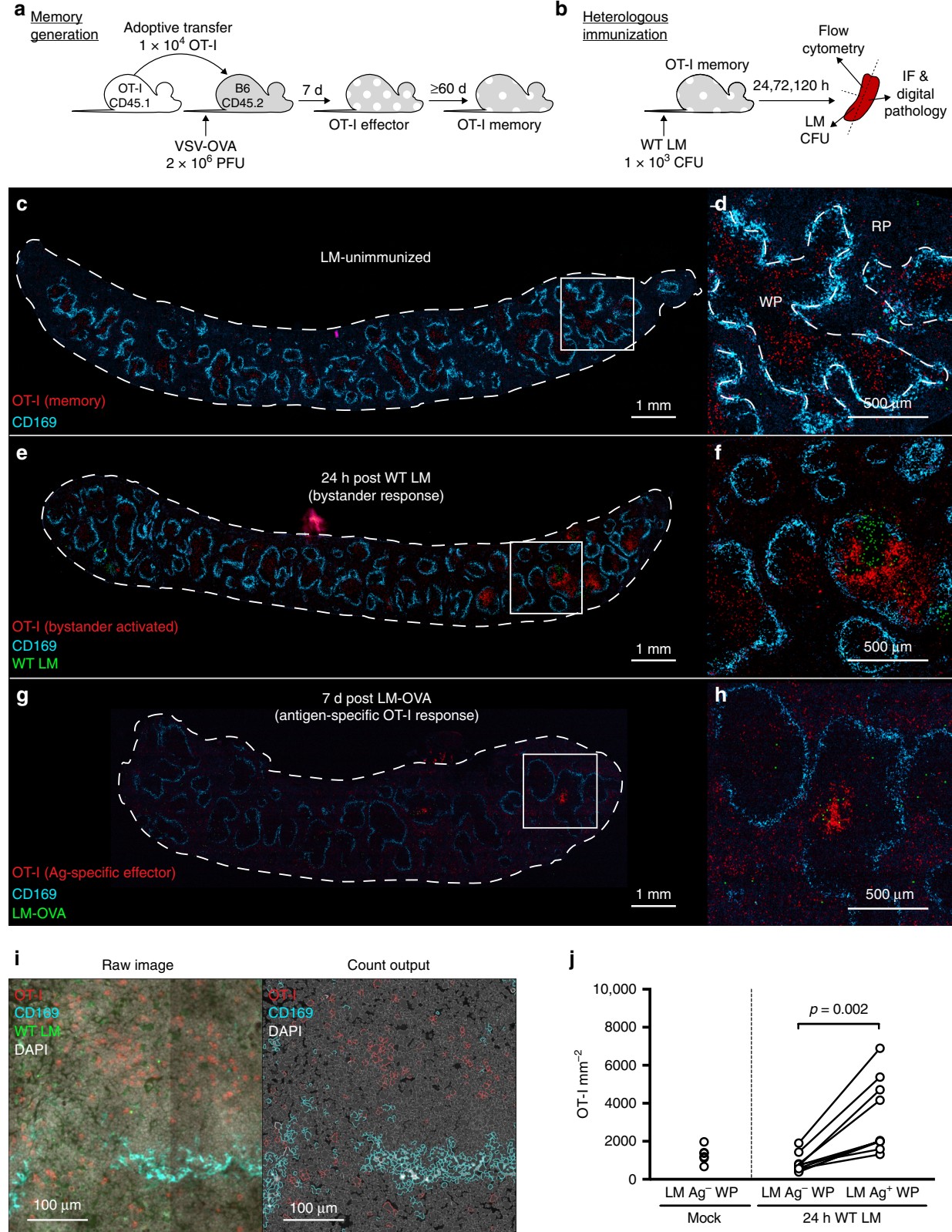

to determine whether bystander activation only occurred passively with memory T cells close to infected/activated APCs or was the result of site-specific recruitment of memory T cells.

We examined the distribution of splenic OT-I T cells in situ using immunofluorescence (IF) of whole-spleen sections (Fig. 1c–h, Supplementary Fig. 2c, d). Using antibodies against CD45.1, Listeriolysin O (LLO), and CD169, we respectively stained for congenically marked OT-I cells, LM Ag, and marginal zone metalophillic macrophages (MMM). CD169+ MMM border the splenic white pulp (WP), delineating areas of WP and red pulp (RP)[30]. Immunization with WT LM led to foci of LM Ag in a limited number of WP in the spleen (Fig. 1e, f) and OT-I T cells formed dense clusters bordering specifically at sites that stained positive for LM Ag. These OT-I T cells appeared to cluster in a

**Fig. 1** Memory CD8+ T cells densely cluster at sites of early immune activation. **a** Schematic of OT-I T cell adoptive transfer and subsequent memory OT-I T cell generation via VSV-OVA infection. **b** Schematic of (bystander-activating) WT LM immunization and subsequent tissue sampling. **c–h** Representative 8 μm, whole-spleen sections showing OT-I (red), MMM (cyan), and LM Ag (green). **c** Whole-spleen section and magnified selection **d** from LM-unimmunized OT-I memory mouse. **e** Whole-spleen section and magnified selection **f** from OT-I memory mouse 24 h post WT LM (bystander-activating) immunization. **g** Whole-spleen section and magnified selection **h** from animal 7 days post OT-I transfer and LM-OVA immunization, showing OT-I effector (Ag-specific) response. **i** Raw IF images showing OT-I (red), MMM (cyan), LM Ag (green), and DAPI (gray), and cell identity outputs used for cell enumeration (OT-I, red; MMM, cyan; co-staining, white; nuclei, gray) from HALO digital pathology software. **j** Splenic OT-I T cell densities from WT LM Ag-positive and -negative WP as enumerated from HALO-analyzed IF images. In **c–h** image contrast of single-channel images was increased using Adobe Photoshop equally across all samples prior to layer compilation. Pixel size for LM Ag channels was doubled to increase visibility using Adobe Photoshop. **c**, **d** Is representative of n = 6. **e**, **f** is representative of n = 10. **g**, **h** is representative of n = 3. In **j** each individual symbol (mock immunization) or connected symbol pair (24 h post WT LM immunization) represents one animal from 3 experiments (n = 6 mock-immunized, n = 10 WT LM-immunized). Indicated are statistical significances by Wilcoxon matched-pairs signed rank test. See also Supplementary Fig. 2. Source data are provided as a Source Data file

manner strikingly similar to day 7 (Ag-specific) effector cells (Fig. 1e–h). We employed a digital pathology software, HALO, to enumerate OT-I T cells within the splenic RP, WP (both as a whole or stratified by LM Ag staining) in an unsupervised manner (Fig. 1i, j). Although WT LM immunization did not globally alter OT-I T cell density from that in unimmunized control animals (Supplementary Fig. 2c), the density of OT-I T cells was significantly increased in WP foci that stained for LM Ag versus those that were negative for LM Ag (Fig. 1j). This suggests that the transient bystander activation of memory T cells (Supplementary Fig. 2e, f) early after immunization is not a consequence of being merely close to activated APCs, but instead memory T cells appeared to actively migrate toward activated APCs. We next wanted to define how this may happen and discern migration from proliferation-driven effects.

**Early bystander-activated T cell clusters are Ki-67 negative.** Using the same experimental setup, we next confirmed that OT-I T cells near splenic LM Ag foci were truly bystander-activated (i.e., granzyme B+) using IF. We simultaneously assayed for the expression of Ki-67, a marker indicative of cellular proliferation, to determine whether the increased density of OT-I T cells resulted from OT-I replication. Within 24 h of WT LM immunization, clustering OT-I T cells predominantly contained cytotoxic molecules and were thus bystander-activated (Fig. 2a–e). Despite this, clustering OT-I T cells did not show Ki-67 staining (Fig. 2a, b), suggesting OT-I T cell enrichment results from recruitment. OT-I T cells from LM-uninfected animals displayed minimal granzyme B staining (Supplementary Fig. 3a). We again employed HALO digital pathology software to enumerate Ki-67+ and granzyme B+ OT-I T cells within WP that stained positive or negative for LM Ag (Fig. 2c). The density of Ki-67+ OT-I T cells remained unchanged between LM Ag-positive and -negative WPs (Fig. 2f); but the density of granzyme B+ cells was highly elevated within WP containing detectable LM Ag (Fig. 2d), with approximately half of the memory OT-I T cells expressing granzyme B within these foci (Fig. 2e). Thus, bystander-activated cells capable of killing are most enriched at sites in close proximity to activated APCs and Ag early after immunization. To ensure that this phenomenon is not limited to TCR transgenic memory T cells, we next sought to determine whether endogenous memory CD8+ T cells are also initially bystander-activated in a locally restricted manner.

**Bystander activation of endogenous memory CD8+ T cells.** As bystander activation occurred in the WP of the spleen, we employed intravenous (i.v.) labeling of CD8β to distinguish OT-I and endogenous CD8+ T cells located in the WP vs. RP of the spleen by flow cytometric analysis (Fig. 3a)[31,32]. In vivo i.v. labeling of CD8β resulted in uniform staining of cells in

circulation and limited staining of cells within lymphoid structures (i.e., lymph node (LN) and splenic WP); as the RP of the spleen is circulatory in nature, we identified CD8β+ splenocytes as those within the RP using both IF and flow cytometry (Fig. 3b, c). From our flow cytometric analyses, we interrogated the expression of effector molecules, granzyme B and IFNγ, in OT-I and bulk endogenous CD8+ T cells from the splenic RP and WP (Fig. 3d, e). Mirroring our IF data, we observed that 24 h after WT LM immunization, a greater frequency of bystander-activated (i.e., granzyme B+) OT-I T cells were detected in the WP versus the RP (Fig. 3e). This bias for bystander-activated OT-I T cells within the WP was present at later time points but peaked 72 h post immunization (Fig. 3e, Supplementary Fig. 4a). In contrast, the granzyme B response of bulk endogenous CD8+ T cells did not mount until 120 h post immunization, likely reflecting the Ag-specific immune response to WT LM (Fig. 3e, Supplementary Fig. 4a). Transient IFNγ production was observed in WP OT-I T cells 24 h post immunization, coinciding with initial OT-I T cell clustering (Fig. 3e, Supplementary Fig. 4b). In concurrence with our IF data, Ki-67 expression was not increased in WP or RP OT-I T cells until later timepoints (Supplementary Fig. 4c, d). To ensure that the bystander responses were not a feature unique to transgenic OT-I T cells, we measured granzyme B and IFNγ responses in endogenous memory CD8+ T cells, which were identified via NKG2D expression (Fig. 3f, g)[33]. Much like OT-I T cells, endogenous memory CD8+ T cells had similar granzyme B and IFNγ kinetics (Fig. 3h, Supplementary Fig. 4a–c). The majority of IFNγ-producing cells within the WP 24 h post immunization, whether OT-I or endogenous memory CD8+ T cells, were predominantly granzyme B+ (Fig. 3g). The magnitude of granzyme B responses 24 h post immunization was directly correlated with the magnitude of the IFNγ responses in both endogenous memory CD8+ and OT-I T cells (Fig. 3i). Furthermore, at 24 h post immunization, the magnitude of the IFNγ and granzyme B responses in endogenous memory CD8+ T cells mirrored the response magnitudes in their OT-I counterparts (Fig. 3j), which remained significant at 72 h post immunization, but not at 120 h post immunization (Supplementary Fig. 4e, f). We found that splenic IFNγ expression 24 h post immunization strongly correlated with splenic IL-12 levels (Fig. 3k, l) and next used IF to determine the magnitude and site-specific manner of IFNγ production. In concordance with our flow cytometry data, IFNγ production was most pronounced in cell clusters surrounding LM Ag foci (Fig. 3m). We leveraged IF staining NKG2D to identify the spatial orientation of endogenous memory CD8+ T cells and NK cells. Endogenous NKG2D+ cells aggregate near LM Ag foci and upregulate granzyme B (Fig. 3n). We next examined if CD69+ CD103+ CD8+ T (resident memory phenotype) cells become similarly bystander-activated during WT LM immunization (Supplementary Fig. 5a, b). We observed an increase in CD69 expression by T cells as expected given that

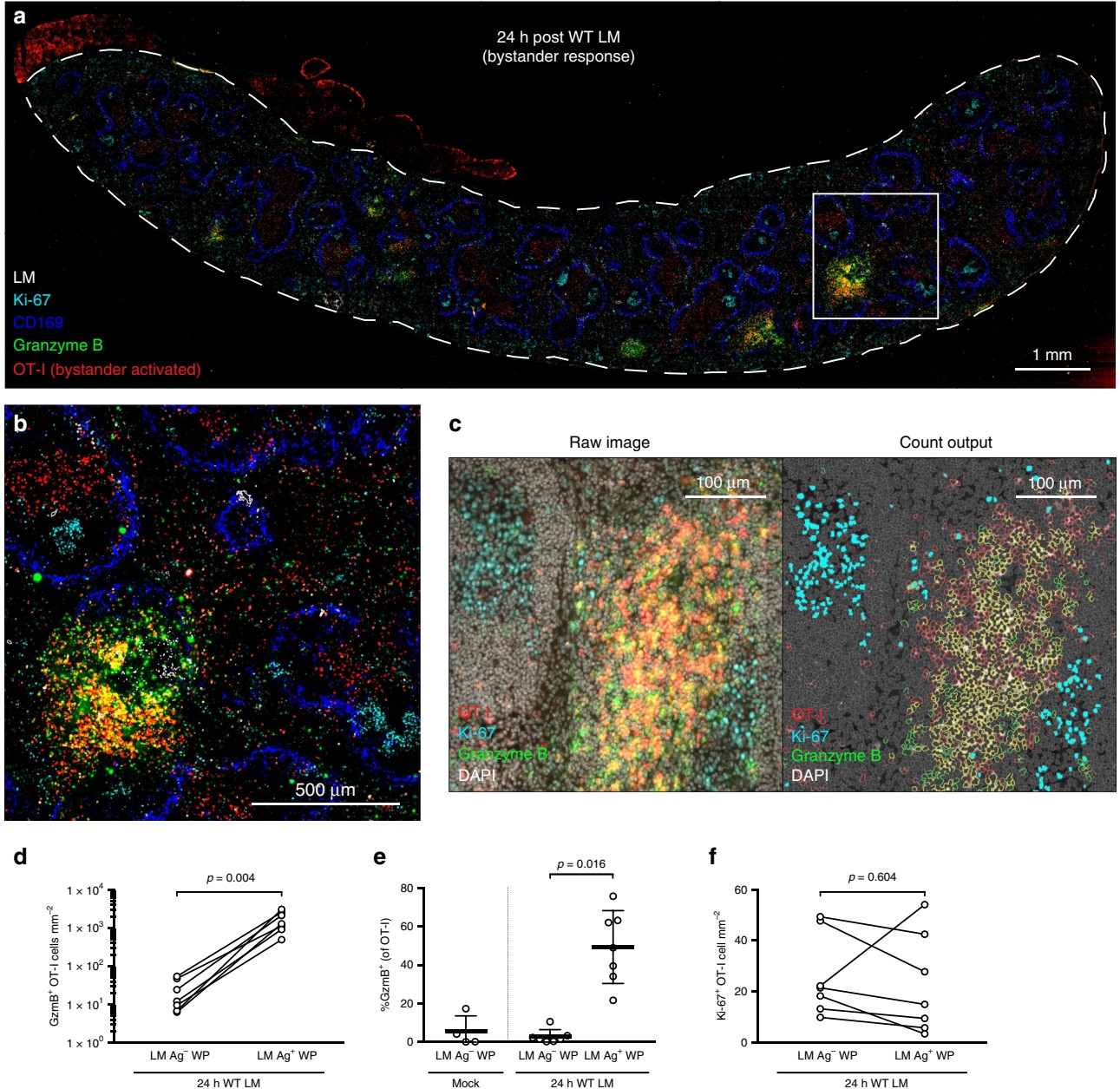

**Fig. 2** Bystander-activated OT-I T cells form clusters without proliferating. **a**, **b** Representative image of spleen 24 h post WT LM immunization from serially sectioned 8 μm slides stained for LM (white), CD169+ MMM (blue), OT-I (red), granzyme B (green), and Ki-67 (cyan). **c** Raw IF images showing OT-I (red), Ki-67 (cyan), granzyme B (green), and DAPI (gray), and cell identity outputs used for cell enumeration (OT-I, red; granzyme B, green; OT-I and granzyme B co-staining, yellow; Ki-67+ nuclei, cyan; Ki-67− nuclei, gray) from HALO digital pathology software. **d–f** Enumeration of cell densities and frequencies from HALO-analyzed IF images. **d** Density of granzyme B+ OT-I cells amongst splenic WP stained for the absence (LM Ag−) or presence (LM Ag+) of WT LM 24 h after immunization. **e** Frequency of granzyme B expression in OT-I T cells within WP from unimmunized animals (Mock) and WP from animals 24 h post WT LM immunization (24 h WT LM), stratified by presence of LM Ag (LM Ag−, LM Ag+). **f** Density of Ki-67+ OT-I T cells amongst splenic WP absent or containing LM Ag 24 h after WT LM immunization. In **a**, **b** image is representative of *n* = 7 spleens. Contrast was increased and tissue orientation was modified in Adobe Photoshop to overlay serially sectioned slides. LM Ag staining was outlined to increase visibility using "find edges" in ImageJ; after, all channels were merged using ImageJ. In **d–f** each symbol (mock immunization) or connected symbol pair (24 h post WT LM immunization) represents one animal from three experiments (*n* = 4 mock-immunized, *n* = 7 WT LM-immunized). Indicated are statistical significances by Wilcoxon matched-pairs signed rank test. See also Supplementary Fig. 3. Source data are provided as a Source Data file

inflammatory signals are sufficient to induce CD69 expression. Importantly, the CD69+ CD103+ population only changed slightly in frequency and showed minimal expression of effector molecules compared with OT-I or endogenous memory CD8+ T cells (Supplementary Fig. 5c–g). These data suggest that bystander-activated cells are primarily migratory and not resident in nature. To better understand how memory T cells are recruited

and activated in this site-specific manner we next wanted to characterize these cells more in depth.

**Surface CXCR3 is decreased on bystander-activated T cells.** As we observed bystander activation of OT-I and endogenous memory CD8+ T cells within the WP 24 h post immunization, we

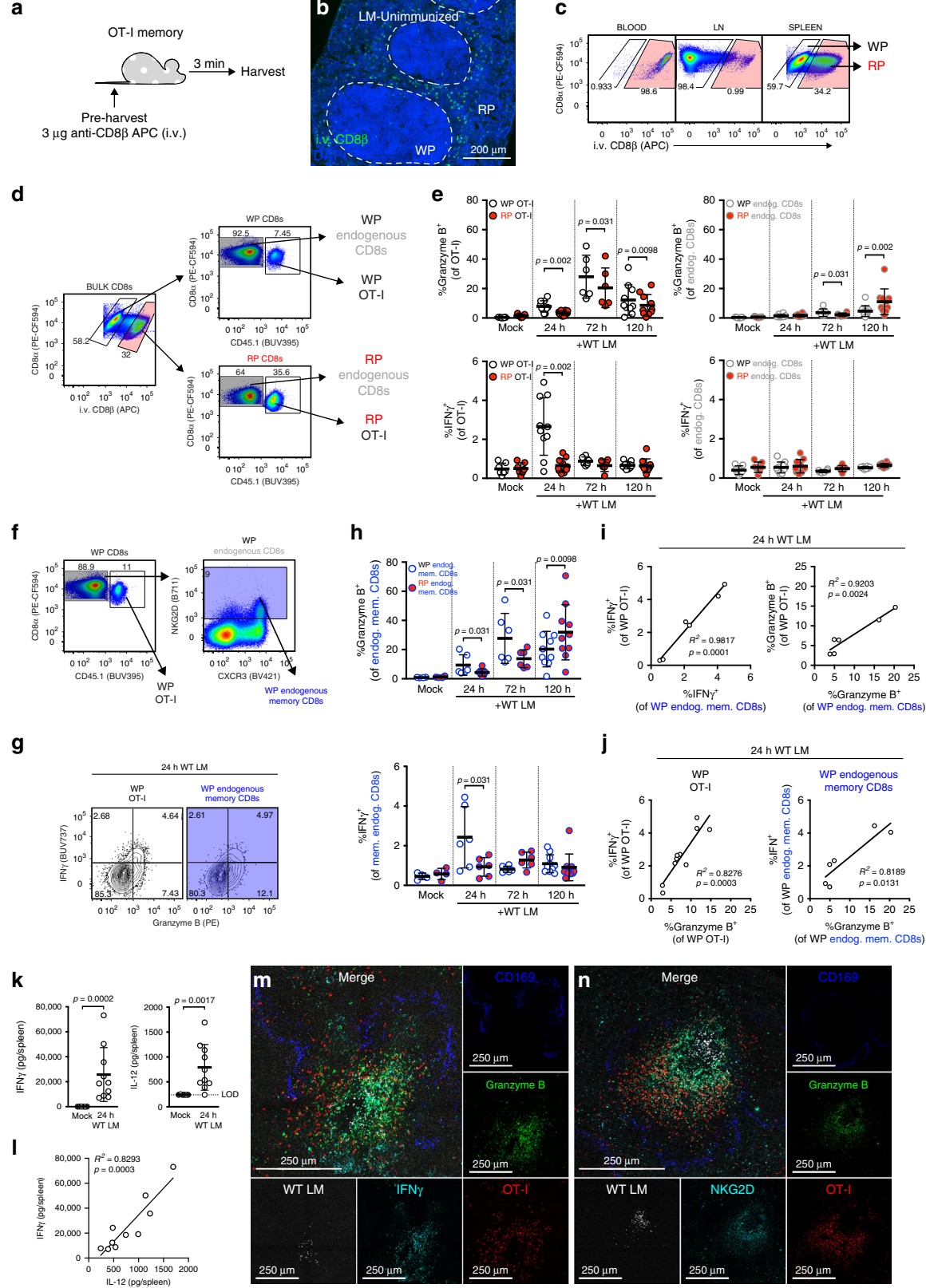

focused our following analyses solely on cells residing in the WP. We sought to determine whether bystander activation at this site is associated with other phenotypic changes. We compared the expression of phenotypic markers between populations of OT-I and endogenous memory CD8⁺ T cells sourced from LM-unimmunized and -immunized (24 h post immunization)

animals (Fig. 4a). We further subset WP OT-I and endogenous memory CD8⁺ T cells from LM-immunized animals by their bystander activation status (i.e., granzyme B positivity) and included a reference population of endogenous naive (NKG2D⁻ CD62L⁺ CD127⁺) CD8⁺ T cells as a baseline control for phenotypic changes (Fig. 4b). An effector (CD62L⁻ CD127⁻)

**Fig. 3** Endogenous memory CD8[+] T cells are bystander-activated in a localized manner. **a** Schematic for i.v. labeling of circulating CD8[+] cells. **b** Splenic IF for i.v.-labeled cells. **c** Representative flow staining of CD8α[+] cells for i.v. CD8β from matched blood, lymph node (LN), and spleen. **d** Representative flow gating for red pulp (RP) and white pulp (WP) subsets within bulk endogenous CD8[+] T cells and OT-I T cells. **e** Frequencies of granzyme B or IFNγ expressing OT-I or bulk endogenous CD8[+] T cells after WT LM immunization, subset by splenic compartment. **f** Representative flow gating for endogenous memory CD8[+] T cells via NKG2D expression. **g** Representative flow plots demonstrating effector molecule production 24 h post immunization in OT-I (white) and endogenous memory CD8[+] T cells (blue) from WP. **h** Frequency of granzyme B[+] and IFNγ[+] endogenous memory CD8[+] T cells within splenic RP and WP after immunization. **i, j** Comparison of **i** effector molecule expression between OT-I and endogenous memory CD8[+] T cells or **j** granzyme B expression versus IFNγ expression 24 h post immunization. **k** Luminex analysis of spleens 24 h after immunization for IFNγ (left) and IL-12p70 (right) and **l** comparison of IFNγ and IL-12p70 levels. **m, n** Representative IF from animals 24 h after immunization, overlaying two serial spleen sections stained for CD169[+] MMM (blue) and WT LM (white); and granzyme B (green), OT-I T cells (red), and **m** IFNγ or **n** NKG2D (cyan). In **e, h** each symbol represents one splenic tissue compartment from one mouse from three experiments (n = 6, 10, 7, and 10, respectively for mock, 24 h, 72 h, and 120 h samples). Indicated are statistical significances by Wilcoxon matched-pairs signed rank test. In **i, j**, each symbol represents one mouse from three experiments (n = 6 and 10). In **k, l**, each symbol represents one mouse from two experiments (n = 6 unimmunized and 10 immunized). Indicated in **k** are statistical significances by Mann–Whitney tests. Summaries shown in **e, h, k** are mean ± SD. Summaries shown in **i, j, l** are linear regressions, with indicated statistical significance of regression fits. **m, n** are representative of n = 2. See also Supplementary Fig. 4, 5. Source data are provided as a Source Data file

---

phenotype was enriched in bystander-activated OT-I and endogenous memory CD8[+] T cells 24 h post immunization with WT LM (Supplementary Fig. 6a, b); but the presence of CD62L[+] KLRG1[−] cells within bystander-activated OT-I T cells suggests that the ability to become bystander-activated is not restricted to a specific memory phenotype (Fig. 4c, Supplementary Fig. 6a, b) Most striking, however, were changes in staining profiles for CXCR3, a chemokine receptor needed to allow Ag-specific effector T cells to find infected target cells[34]. In unimmunized animals, memory OT-I and endogenous memory CD8[+] T cells uniformly express CXCR3 (Fig. 4c, d), but within 24 h of WT LM immunization, bystander-activated OT-I and endogenous memory CD8[+] T cells displayed a significant drop in numbers, frequency and MedFI for surface CXCR3 (Fig. 4c, d, Supplementary Fig. 6c). As WT LM detection causes rapid CXCR3 ligand secretion (CXCL9 and CXCL10) by human PBMC[35] and CXCR3 ligand engagement and signaling can lead to dimmed staining owing to receptor internalization[36], we next asked if this loss of surface CXCR3 expression we observed could be due to ligand engagement at sites of early immune activation. We isolated OT-I and endogenous memory CD8[+] T cells and exposed them to recombinant CXCL9 and CXCL10 (Fig. 5a, b). We observed a rapid, CXCL9 and CXCL10 dose-dependent decrease in CXCR3 expression on both OT-I and endogenous memory CD8[+] T cells (Fig. 5c, d, Supplementary Fig. 7a) and cycloheximide (CHX)-independent CXCR3 re-expression following ligand removal (Supplementary Fig. 7b-e). These data indicate that murine CXCR3 can be recycled to the cell surface following internalization, whereas human CXCR3 is restored via de novo protein synthesis[37]. Finally, we wanted to define CXCL9 and CXCL10 expression in situ and stained serially sectioned slides for CD169 and LLO; CXCR3L (CXCL9 and CXCL10); and CD45.1 (for OT-I T cells) and CD11b. We found that WP OT-I T cells colocalized in areas staining for both CXCL9 and CXCL10 (Fig. 5e, f, Supplementary Fig. 7f). Thus, we next asked if CXCR3 is necessary for recruitment of memory T cells to activated APCs and undergoing bystander activation.

**Localized bystander activation requires CXCR3.** To determine whether recruitment of memory T cells towards activated APCs is CXCR3-dependent, we generated memory mice using WT or Cxcr3[−/−] OT-I T cells (as described in Fig. 2a) and immunized the mice with WT LM to induce bystander activation (Fig. 6a). We interrogated the bystander responses in the WP of OT-I (either WT or Cxcr3[−/−]) and endogenous memory CD8[+] T cells (which are all Cxcr3[+/+]) (Fig. 6b). After WT LM immunization, Cxcr3[−/−] OT-I T cells had limited IFNγ and granzyme B

expression in comparison with WT OT-I and endogenous memory CD8[+] T cells (from both WT or Cxcr3[−/−] OT-I T cell transfers) (Fig. 6c). At 24 h post immunization, the frequency of IFNγ[+] or granzyme B[+] Cxcr3[−/−] OT-I T cells was significantly less than endogenous memory CD8[+] T cells from the same animal (Fig. 6d) or WT OT-I T cells from animals of the experimental control group (Fig. 6e, f). Although there were pronounced CXCR3-dependent differences in IFNγ and granzyme B responses in OT-I T cells, the endogenous memory CD8[+] T cells had IFNγ and granzyme B responses of similar magnitude (Fig. 6e, f). We utilized IF to determine whether clustering near LM Ag and target cells was perturbed by CXCR3-deficiency. Cxcr3[−/−] OT-I memory T cells remained at a higher frequency (as % of CD8[+] T cells) than their WT counterparts during the memory stage but had a stable memory phenotype (Supplementary Fig. 8a). This is in agreement with a previous report illustrating that Cxcr3[−/−] CD8[+] T cells undergo limited contraction but maintain their ability to reactivate and exert cytotoxicity[38]. Nevertheless, the splenic distribution of Cxcr3[−/−] and WT OT-I cells was of a similar pattern (Fig. 6h, mock); this too, was reflected in uninfected WP foci 24 h after WT LM immunization. In contrast to the dense clusters that WT OT-I T cells form with other granzyme B[+] cells around areas rich in LM Ag, Cxcr3[−/−] OT-I T cells remained largely granzyme B[−] and on the periphery of clustering, granzyme B[+] cells (Fig. 6h, 24 h WT LM). Importantly, Cxcr3[−/−] OT-I T cells are not inherently defective in becoming bystander-activated as we observed granzyme B expression at later time points when the infection had become systemic (Fig. 6g). Similarly, Cxcr3[−/−] OT-I memory T cells express IFNγ following in vitro exposure to IL-12, IL-15, and IL-18 (Fig. 6i–m; Supplementary Fig. 8b, c).

Therefore, CXCR3 signaling mediates the chemoattraction of memory T cells to areas with infected/activated APCs. Finally, CXCR3-mediated recruitment is also used by Ag-specific CD8[+] T cells to locate infected target cells[34]. Given this overlap in recruitment mechanism, we hypothesized that bystander-activated CD8[+] T cells can temporospatially overlap with Ag-specific CD8[+] T cells. Using adoptive transfers of P14 CD8[+] T cells (a TCR transgenic which recognizes lymphocytic choriomeningitis virus Ag, gp33), we interrogated how bystander-activated OT-I T cells were distributed in comparison with Ag-specific P14 T cells after LM-gp33 immunization (Supplementary Fig. 8d). Ag-specific P14 T cells were detectable in the spleen 96 h post immunization; however, P14 T cells formed discrete clusters that were not integrated with clusters of OT-I T cells, granzyme B[+] endogenous cells, and LM Ag (Supplementary Fig. 8e).

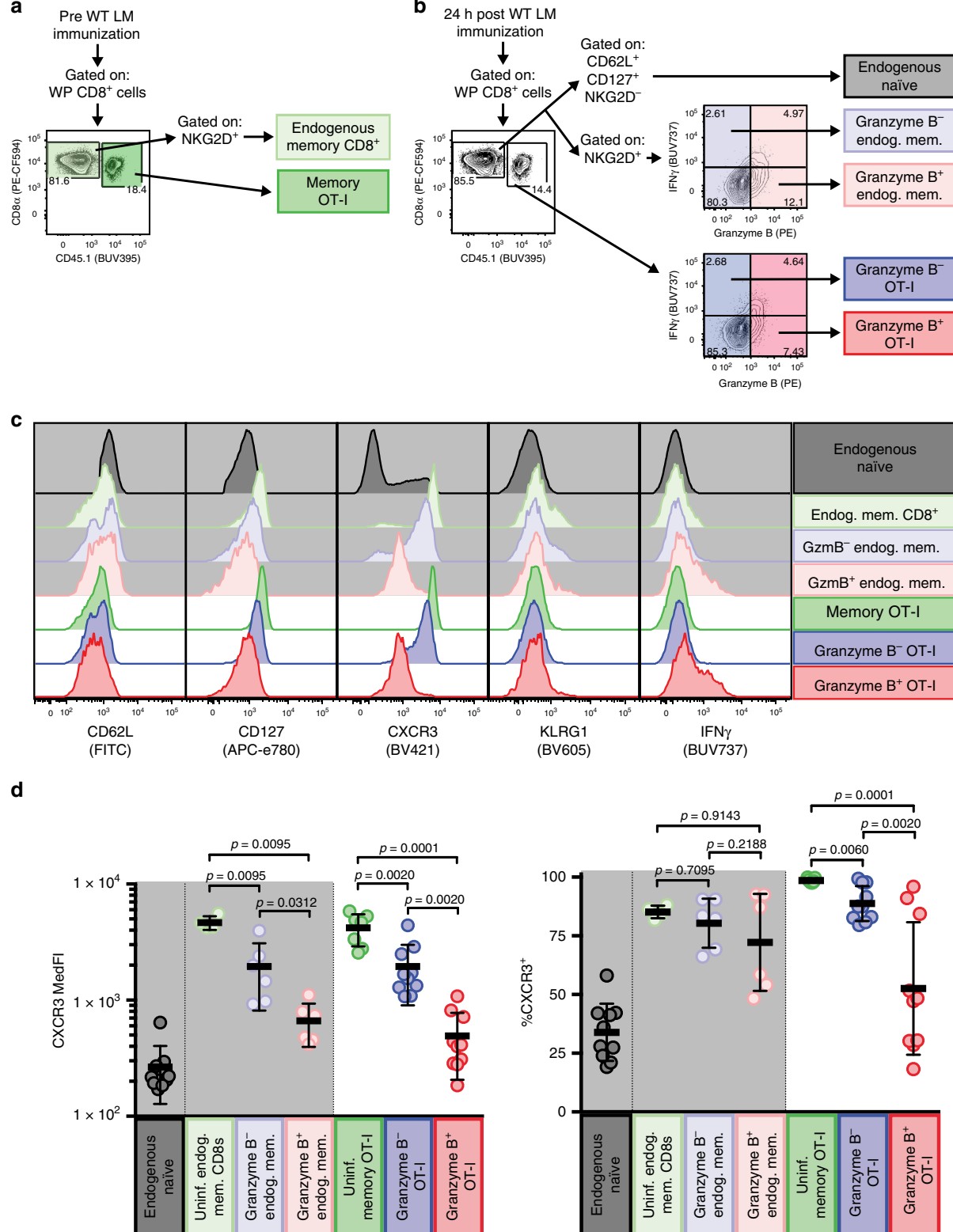

## Discussion

Bystander activation of memory T cells has been reported in the context of infections that are systemic in nature or have systemic inflammatory effects[10]. In these instances, memory T cells are exposed to systemic inflammatory cues and essentially become bystander-activated in a passive manner. We initially asked if bystander activation occurs when the inflammatory event is localized, specifically following delivery of an intra-muscular vaccine. Three days post vaccination, we observed phenotypic changes in the CD8[+] T cell compartment in vaccine recipients (but not in placebo recipients) that were consistent with bystander activation (Supplementary Fig. 1). Based on these data, we concluded that the i.m. vaccine either elicited a systemic inflammatory immune response strong enough to induce bystander activation or, alternatively, it could indicate that there was localized bystander activation in the draining lymph nodes

**Fig. 4** Bystander-activated T cells display a decrease in CXCR3 cell surface expression. **a, b** Representative flow plots showing gated populations used for representative overlays in **c** and frequencies and median fluorescence intensity (MedFI) reported in **d**. **a** Representative flow plot from LM-uninfected OT-I memory white pulp (WP) splenocytes showing gating strategy for endogenous memory CD8[+] T cells (light green) and memory OT-I T cells (green). **b** Representative flow plot from OT-I memory WP splenocytes 24 h post WT LM infection, showing gating strategy for endogenous naive CD8[+] T cells (gray), granzyme B[+] (i.e., bystander-activated) and granzyme B[−] endogenous memory CD8[+] T cells (light red and light blue, respectively) and OT-I T cells (red and blue, respectively). **c** Flow histogram overlays of populations outlined in **a**, **b** for phenotypic and functional markers. Cells endogenous to hosts have a light gray background. **d** CXCR3 MedFI and CXCR3[+] frequencies of populations outlined in **a**, **b**. **b** is a representative plot from $n = 3$ animals concatenated into a single file. **c** is a representative plot from concatenated populations shown in **a**, **b**. In **d** a single point represents a specific population from each animal ($n = 6$ for endogenous memory populations and 10 for transgenic and endogenous naive populations) from three experiments. Summary statistics shown are mean ± SD. Indicated are statistical significances by Wilcoxon matched-pairs signed rank test. See also Supplemental Fig. 6. Source data are provided as a Source Data file

followed by release of the bystander-activated T cells to the periphery. As we cannot experimentally test if human memory T cells became bystander-activated in the lymph node draining the vaccination site, we used a mouse model to determine whether localized bystander activation could occur. We chose a low-dose LM mouse model of immunization in which activated APCs and inflammation are initially constrained to a limited number of white pulps in the spleen (Fig. 1d). We found that memory OT-I T cells were specifically recruited to and enriched at sites of early immune activation and bystander-activated in a highly localized yet still rapid manner (Figs. 1, 2). These data strongly suggested the existence of a mechanism that allowed active migration to the site of inflammation at a timepoint (24 h post immunization) that is typically considered to be dominated by the innate immune response[39]. Previous reports also demonstrated early bystander-mediated activation of memory T cells, but it is important to consider that these previous studies immunized/infected mice with high doses of attenuated $actA^−$ or WT LM, resulting in ubiquitous infection instead of localized infection[6,8]. These ubiquitous infection models have proven highly valuable to understand the inflammatory signals that activate memory T cells, but they cannot determine whether bystander activation is merely a passive process. The strength of our low-dose WT LM infection model is that immune cell activation remains anatomically confined within the first 24 h of immunization. Furthermore, the early decrease of circulating OT-I and CD8[+] T cells 24 h after WT LM immunization mirrors the decrease of circulating human CD8[+] T cells after i.m. immunization with recombinant vectors[40].

Thorough characterization of the bystander-activated T cells revealed a decrease of CXCR3 expression levels specifically on granzyme B[+] T cells indicating that these cells may have received a CXCR3-mediated signal leading to receptor internalization, which was supported by colocalization of CXCR3L and OT-I T cells in situ (Figs. 4, 5). Using $Cxcr3^{−/−}$ OT-I T cells, we found that CXCR3 was indeed necessary to recruit memory T cells, which clustered tightly around these APCs (Fig. 6) and became bystander-activated, including expression of granzyme B and IFNγ (Fig. 6). Importantly, since even the low-dose LM infection will spread over time and result in a systemic infection, we could examine if $Cxcr3^{−/−}$ OT-I T cells became bystander-activated once the infection had become systemic. We found that $Cxcr3^{−/−}$ OT-I T cells were bystander-activated at a later timepoint and thus ruled out that $Cxcr3^{−/−}$ memory T cells have an inherent functional defect that rendered them incapable to become bystander-activated at all (Fig. 6, Supplementary Fig. 7B). Together, these data support a model where CXCR3-mediated signals drive memory T cells to sites of inflammation to become bystander-activated.

Using a microscopy approach, a previous study showed that memory T cells spatially orient themselves in areas prone to early immune activation within the lymph node, predisposing them to

bystander activation and facilitating rapid responses[41]. Using unsupervised digital pathology software analysis, our data demonstrate that in addition to this mechanism that predisposes memory T cells to spatially encounter inflammation, memory T cells also possess the ability to be recruited to sites of inflammation and participate in early immune responses (Figs. 2, 3). The subsequent immunological consequences of bystander activation are likely highly context-dependent and even the initial trigger for bystander activation can vary substantially[10]. In our mouse model, we elicited a localized infection in animals that do not possess antigen-specific memory T cells. Masopust et al.[42] exposed tissue-resident antigen-specific T cells to their cognate antigen and showed that this could also induce recruitment of bystander-activated T cells into the area where antigen was provided using a different, IFNγ-dependent mechanism. Considering that bystander activation occurs in all these different experimental systems, we argue that bystander activation is not an immunological accident or vestigial feature of memory T cells, but a well-orchestrated and conserved part of the immune response. We previously suggested that once bystander-activated, memory CD8[+] T cells functionally resemble NK cells[7]. Both are capable of killing targets upon engagement of innate immunoreceptors (specifically NKG2D); however, bystander-activated memory CD8[+] T cells typically do not express innate inhibitory receptors that restrict killing[7]. In addition, both migrate to sites of early immune activation and produce IFNγ, although NK cell migration appeared to be predominantly dependent on CCR5[39]. Thus, bystander-activated Ag-nonspecific CD8[+] T cells can execute an NK-like program without being susceptible to inhibitory signals capable of disrupting NK-mediated killing/migration. An elegant study by Teixeiro and colleagues demonstrated that memory T cells require tonic recognition of self (peptide/MHC) (beneath the threshold of TCR binding to cognate Ag/MHC) to become bystander-activated[43], indicating that the NK cells and bystander-activated T cells have complimentary rather than congruent surveillance functions despite the functional overlap. The goal is likely the same—curtail initial pathogen spread until the adaptive immune response is fully developed.

Finally, our data also provide a plausible explanation for why tumor nonspecific memory T cells are part of the T cell infiltrate within solid tumors[18]. Given that a tumor with a T cell infiltrate is by definition inflammatory and our data show that memory T cells migrate to sites of inflammation, memory T cells may be continually recruited in this context. Bystander-activated T cells do not see antigen in the tumor and should not become functionally exhausted, as exhaustion is a TCR-dependent process[44], which is also consistent with their recently reported phenotype in tumors[18]. Bystander-activated T cells could potentially contribute to tumor clearance, either by NKG2D—NKG2D ligand mediated cytotoxicity or by IFNγ-mediated macrophage activation, which has been shown to lead to tumor cell clearance when NK cells were the source of IFNγ[45]. However, these bystander-activated

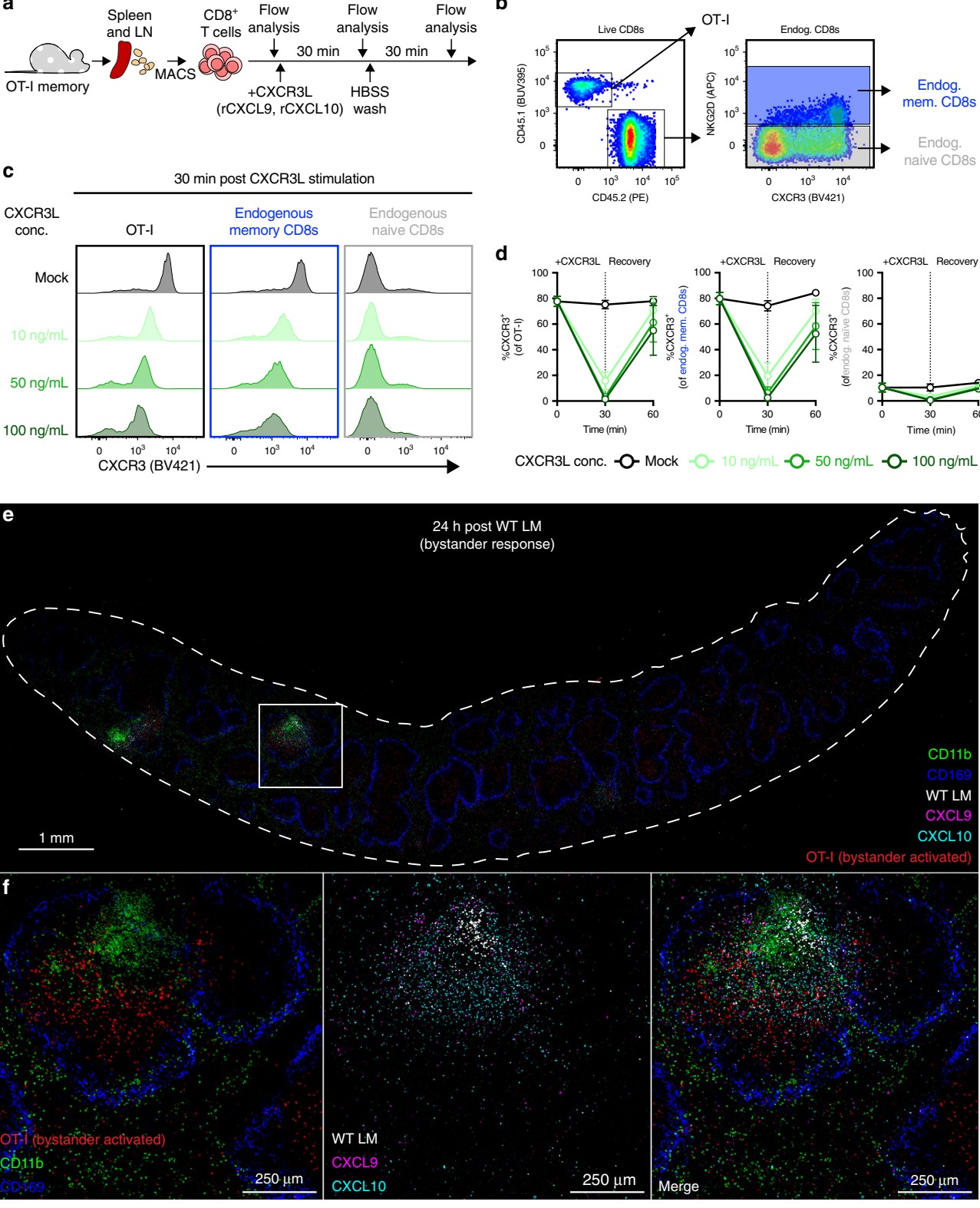

T cells may also occlude antigen-specific T cells, as both bystander-activated T cells and antigen-specific effector cells use CXCR3 to find target cells[34], thus competing for the same signals. Two recent studies demonstrate that the naive T cell response is highly sensitive to competition for antigen and even a minimal overlap in epitope-specificity by a memory T cell population is sufficient to impair a primary T cell response[46,47]. Future studies will need to address if bystander-activated T cells are similarly effective at impairing primary T cell responses, particularly lower affinity responders[48], by limiting antigen availability. This is particularly relevant in the context of vaccination as we found that bystander-activated T cells still surrounded APCs as the

**Fig. 5** CXCR3 ligands are expressed at sites of early immune activation. **a** Schematic illustrating CD8+ T cell isolation, CXCR3 ligand (CXCR3L) stimulation, and longitudinal sampling for flow cytometric analysis. **b** Representative gating of OT-I T cells, NKG2D+ endogenous memory CD8+ T cells (blue), and NKG2D− endogenous naive CD8+ T cells (gray). **c** Representative histograms depicting surface CXCR3 flow staining in OT-I T cells (black border), endogenous memory CD8+ T cells (blue border), and endogenous naive CD8+ T cells (gray border) after 30 min of mock (dark gray histogram) or CXCL9 plus CXCL10 (rCXCR3L) stimulation (10 ng/mL, light green; 50 ng/mL, medium green; 100 ng/mL, dark green per chemokine). **d** Frequencies of OT-I T cells (left), endogenous memory CD8+ T cells (center), and endogenous naive CD8+ T cells (right) expressing surface CXCR3 prior to rCXCR3L stimulation (0 min), after rCXCR3L stimulation (30 min), and 30 min after rCXCR3L removal (60 min). Symbols and line connectors are color-coded by rCXCR3L concentration as in **c**. **e**, **f** Representative IF image of spleen from animal 24 h post WT LM immunization. The image is composed of three 8 μm serial sections stained for CD11b (green) and CD169+ MMM (blue); OT-I T cells (red) and WT LM (white); and CXCL9 (magenta) and CXCL10 (cyan). **c** is a representative plot from one of two technical replicates. In **d** each symbol represents the mean ($n = 2$ at each timepoint) ± SD with lines connecting means from two technical replicates. **e**, **f** is representative of $n = 5$ animals from two technical replicates. Image orientation was modified in Adobe Photoshop to permit overlaying of serially sectioned slides. Contrast was equally increased amongst all individual images; after, punctate staining in LM, CXCL9, and CXCL10 images was outlined using ImageJ command "find edges" to enhance visibility prior to merging. See also Supplementary Fig. 7. Source data are provided as a Source Data file

antigen-specific effector T cells developed and entered the area. Future studies will also need to examine if there is competition between antigen-specific effector cells and bystander-activated memory T cells and determine whether effector cells have other means to outcompete bystander-activated memory T cells to access target cells or APCs. These interactions should also be considered for IL-15-based immunotherapies[49] that would activate antigen-specific and nonspecific T cells.

In summary, based on our data we propose that memory T cells, a hallmark population of the adaptive immune system, play an active and critical role in the initiation of the early immune response even when inflammatory events are highly localized.

## Methods

**Human clinical samples from HVTN 908**. We acquired cryopreserved PBMC from participants enrolled in HVTN 908 (ClinicalTrials.gov NCT00908323), a parallel sub study of the HIV vaccine trial, HVTN 205 (ClinicalTrials.gov NCT00820846), in which volunteers were vaccinated with pGA2/JS7 DNA and MVA/HIV62[19]. All participants provided informed consent under HVTN protocols 908 and 205 under approval of HIV Vaccine Trials Network Institutional Review Boards. Our use of these human PBMC samples were approved by the HIV Vaccine Trials Network Institutional Review Boards and we conducted this study in compliance with all relevant ethical regulations. We specifically interrogated PBMC at D112 (pre-immunization with MVA/HIV62) and D115 (3d post immunization with MVA/HIV62), collected under the HVTN 908 protocol. At D112, after PBMC draw, volunteers either received their first MVA/HIV62 boost or a saline placebo. Of these, we received PBMC from $n = 6$ MVA/HIV62 vaccinees and $n = 6$ saline placebo controls. Of note, vaccination regimen was blinded until completion of data acquisition and analysis.

**Flow cytometric analysis of HVTN 908 samples**. We thawed all D112 and D115 samples and immediately stained for markers outlined in Supplementary Table 1. All stains occurred at room temperature (RT). We conducted viability staining in 1× phosphate-buffered saline (PBS) for 20 min. We simultaneously blocked nonspecific Fc and stained for tetramer for 1 h in 2% FBS-supplemented PBS (FACSWash). We then stained all other surface markers in FACSWash for 20 min. After, we fixed cells for 20 min in cytofix/cytoperm (BD Biosciences, San Diego, CA). Afterwards, we stained intracellular markers in 1× perm/wash buffer (BD Biosciences) for 30 min. We resuspended cells in FACSWash before acquiring events on a FACSSymphony (BD Biosciences), which were later analyzed using FlowJo v9 and v10 (TreeStar Inc, Portland, OR). Flow reagents, dilutions, and fixation/permeabilization methods are listed in Supplementary Table 1; gating strategies are depicted in Supplementary Fig. 9.

**Mice**. All mouse protocols in this study were approved by the Fred Hutchinson Cancer Research Center's Institutional Animal Care and Use Committee. All experimentation that we conducted in this study was in compliance with the ethical regulations outlined by the Institutional Animal Care and Use Committee. All mice were maintained in specific pathogen-free conditions. We purchased 6-week-old female C57BL/6 J mice from the Jackson Laboratory (Bar Harbor, ME). We maintained WT OT-I TCR transgenic mice on a CD45.1 background. Dr. Ross Kedl (University of Colorado, Denver) kindly provided $Cxcr3^{−/−}$ OT-I TCR transgenic mice on a CD45.1/CD45.2 background. Dr. Surojit Sarkar (Seattle Children's Research Institute) kindly provided LNs from P14 TCR transgenic mice on a Thy1.1 background. We collected submandibular blood in heparin from facial

vein puncture immediately preceding euthanasia. After euthanasia, via $CO_2$ overdose and cervical dislocation, we collected spleens and axial, inguinal, and cervical LNs from mice and separated these tissues for analyses.

**Viral and bacterial infections/immunizations**. We used an OVA-expressing VSV construct. We used WT and $actA^−$ LM, as well as OVA- and gp33-expressing recombinants of these LM strains. We grew LM in BHI media (Thermo Fisher, Waltham, MA) to the log phase of growth (determined by OD600). All pathogens were diluted in sterile 1× PBS for i.v. injection.

**Adoptive transfers**. We mechanically isolated lymphocytes from LN that were harvested from infection- and immunization-naive, 6- to 12-week-old female OT-I mice. To accomplish this, we mechanically passed tissue through a 70 μm cell strainer. We immediately enriched CD8+ OT-I or P14 T cells using Miltenyi CD8 negative selection antibody cocktail and beads (Miltenyi Biotec, Germany), which we confirmed using flow cytometry. We diluted OT-I T cells in sterile 1× PBS and adoptively transferred $1 \times 10^4$ cells per C57BL/6 J recipient via i.v. injection. In order to visualize Ag-specific effector responses during LM infection, we immunized WT OT-I recipient mice with $2 \times 10^3$ CFU LM-OVA immediately after adoptive transfer. We sacrificed these mice 7 days post LM-OVA immunization. To develop OT-I memory mice, we infected OT-I recipient mice with $2 \times 10^6$ PFU VSV-OVA within 1 day after OT-I adoptive transfer. We surveyed OT-I expansion in peripheral blood at 7 days post immunization to confirm successful VSV-OVA infection. We aged mice ≥ 60 days until contraction of OT-I cells stabilized before conducting bystander-activating LM immunizations. To induce bystander activation in OT-I memory mice, we immunized animals with $1 \times 10^6$ CFU $actA^−$ LM or $1 \times 10^3$ CFU WT LM. We sacrificed mice 24, 72, and 120 h post immunization via $CO_2$ euthanasia.

**In vivo CD8β labeling**. As described[31,32], we i.v. injected mice with 3 μg APC-conjugated anti-CD8β (eBioH35-17.2), which stains cells in circulation (including splenic RP), but not those in parenchymal tissues (including LN and splenic WP). After 3 min, we collected submandibular blood via facial vein puncture, euthanized mice, and harvested tissues. We immediately prepared single-cell suspensions of tissues used for flow cytometric analysis and stained with a distinct anti-CD8α antibody to delineate RP and WP CD8+ T cells (alongside other antibodies for immunophenotyping). We confirmed in vivo CD8β labeling using IF. We stained acetone-fixed 8 μm spleen sections with biotinylated anti-APC (50 μg/mL, 1 h, RT, 5% mouse serum, 5% human serum in TBS). We then stained with streptavidin (SA)-Alexa fluor (AF)647 (20 μg/mL, 1 h, RT, 5% mouse serum, 5% human serum in TBS). We counterstained with DAPI (5 min, RT, PBS) and mounted slides in Prolong Gold (Thermo Fisher). We acquired images on an Aperio SlideScan FL (Leica Biosystems, Wetzlar, Germany), which were then processed in ImageJ (NIH, Bethesda, MD).

**Ex vivo flow cytometric analysis of mouse tissues**. We prepared single-cell suspensions of LN and spleen by mechanically passing tissue through a 70 μm cell strainer in complete RP10 media (RPMI 1640 supplemented with 10% FBS, 2 mM L-glutamine, 100 U/mL penicillin–streptomycin, 1 mM HEPES, 1 mM sodium pyruvate, 0.05 mM β-mercaptoethanol). To count cells, we stained suspension aliquots with trypan blue (Gibco) and counted using a TC20 Automated Cell Counter (Bio-Rad, Hercules, CA). We incubated blood with 1 mL and LN and spleen single-cell suspensions with an equal volume of ACK lysis buffer (Gibco) on ice for 20 min. We conducted all stains for flow cytometry on ice. We conducted viability staining in 1× PBS for 20 min. We stained surface markers in FACSWash + (1× PBS supplemented with 2% FBS plus 0.2% sodium azide, and 2 mM EDTA) for 30 min. We fixed cells in cytofix/cytoperm (BD Biosciences) or eBioscience FOXP3 fixation/permeabilization buffer (Thermo Fisher) for 20 min. We then stained

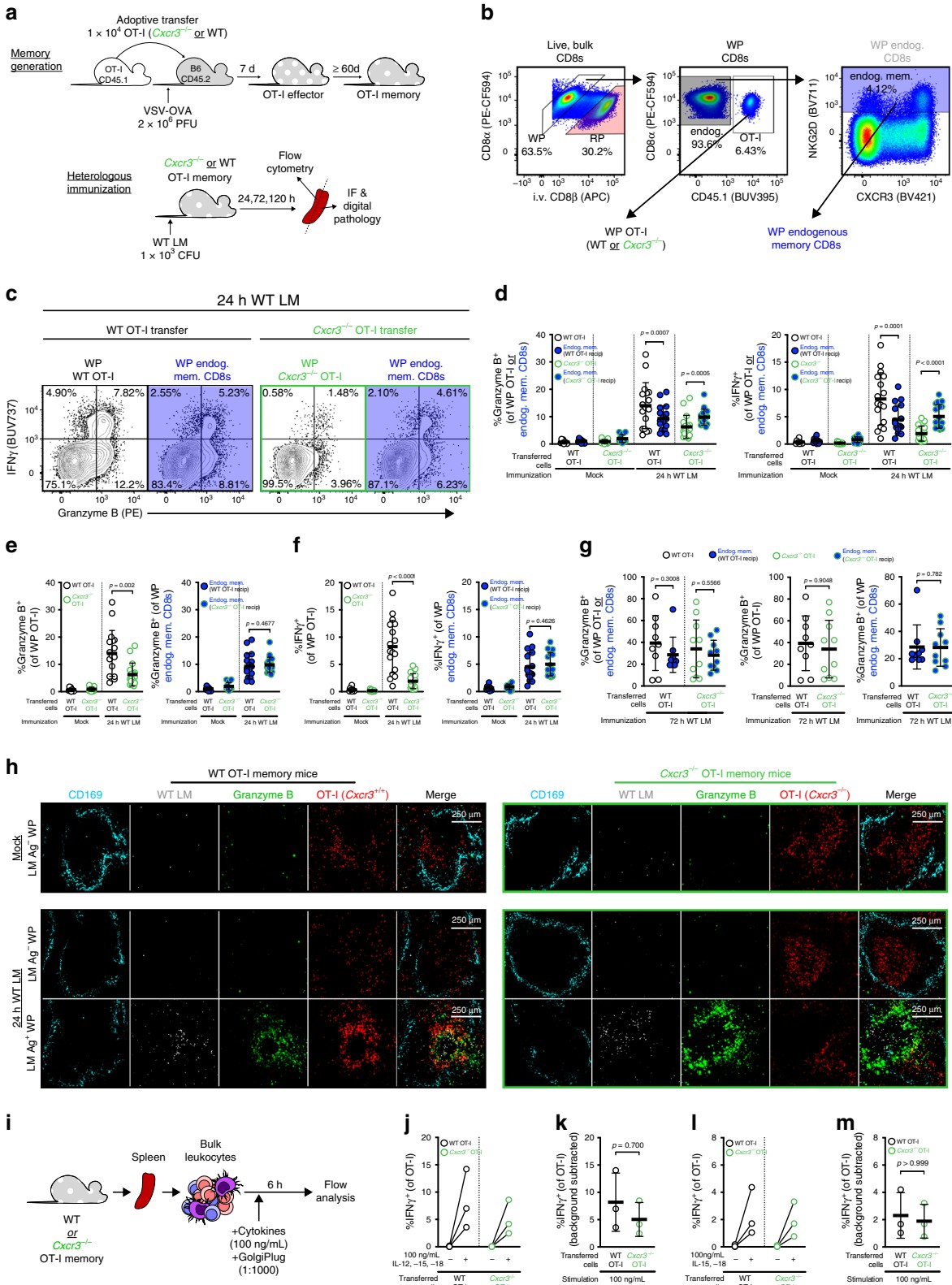

intracellular markers in 1× perm/wash buffer (BD Biosciences) or 1× eBioscience FOXP3 permeabilization buffer (Thermo Fisher) for 30 min. We resuspended cells in FACSWash + before acquiring events on an LSR II (BD Biosciences) or FACSymphony (BD Biosciences), which were later analyzed using FlowJo v9 and v10 (TreeStar Inc, Portland, OR). Flow reagents, dilutions, and fixation/permeabilization methods are listed in Supplementary Table 2. Gating strategy is depicted in Supplementary Fig. 10.

**In vitro CXCR3L stimulations and CHX chase**. We harvested spleen and LN from OT-I memory animals (Fig. 2a) and prepared single-cell suspensions. We immediately enriched CD8+ OT-I using Miltenyi CD8-negative selection antibody cocktail and beads (Miltenyi Biotec). After manually counting cells using trypan blue staining, we plated 0.5–2 million CD8+ T cells per well in a 96-well U-bottom tissue culture plate. We briefly treated cells with sterile dimethyl sulfoxide (DMSO) or 15 µg/mL cycloheximide (CHX) (Sigma Aldrich, St. Louis, MO) as described in

**Fig. 6** Localized bystander activation requires CXCR3-mediated chemotaxis. **a** Experimental overview. **b** Representative gating of WP OT-I and endogenous memory T cell populations. In symbols for **c**–**g**, OT-I and endogenous memory CD8$^+$ T cells are respectively filled white and blue; WT or $Cxcr3^{-/-}$ OT-I T cell recipients are, respectively, outlined in black and green. **c** Representative gating depicting IFNγ and granzyme B expression 24 h post immunization. **d**–**f** Frequency of granzyme B and IFNγ-expressing cell subsets 24 h after immunization. **g** Frequency of granzyme B expression in OT-I and endogenous memory CD8$^+$ T cell subsets 72 h after immunization **h** Representative WP image from WT LM-unimmunized animals (mock, top row) and -immunized animals (24 h post immunization, bottom two rows). Slides were stained for geographic markers: LM Ag (white) and CD169$^+$ MMM (cyan); and phenotyping: OT-I (red) and granzyme B (green). Image borders delineate OT-I genotype (WT, black; $Cxcr3^{-/-}$, green). In lower two rows, LM Ag-negative (top) and Ag-positive (bottom) foci are from the same LM-immunized animal. **i** Stimulation schematic of OT-I T cells with bystander-activating cytokines. **j**, **l** Frequencies of IFNγ-expressing OT-I T cells after mock or cytokine stimulation. **k**, **m** Background-subtracted IFNγ$^+$ frequencies after cytokine stimulation of WT and $Cxcr3^{-/-}$ OT-I T cells. In **d**–**g**, each symbol represents the transgenic or endogenous memory compartment of CD8$^+$ T cells per animal ($n = 9$ mock WT recipients, $n = 12$ mock $Cxcr3^{-/-}$ recipients, $n = 17$ 24-hour WT and $Cxcr3^{-/-}$ recipients, $n = 9$ and 10 72-hour WT and $Cxcr3^{-/-}$ recipients, respectively) across four to six technical replicates. Statistical significance is indicated (**d**, **g** left panel, and **e**, **f**, **g** center and right panels) and was assessed by Wilcoxon matched-pairs signed rank test and Mann–Whitney tests, respectively. In **h**, images are representative of $n = 3$ and 5 (unimmunized WT and $Cxcr3^{-/-}$ OT-I T cell recipients, respectively) and $n = 7$ and 8 (LM-immunized WT and $Cxcr3^{-/-}$ OT-I T cell recipients, respectively). In **j**–**m**, each symbol or connected pair represents one animal ($n = 3$) across three technical replicates. In **d**–**g**, **k**, **m**, summary statistics shown are mean ± SD; indicated are statistical significances by Mann–Whitney tests. See also Supplementary Fig. 8. Source data are provided as a Source Data file

Meiser et al.[37], before stimulating with recombinant mouse CXCL9 and CXCL10 (BioLegend, San Diego, CA) in combination at 10, 50, or 100 ng/mL in complete RP10. Media alone was used as a negative control. We cultured cells in recombinant CXCR3L-containing media at 37 °C 5% $CO_2$ for 30 min. Afterwards, we washed cells three times with 37 °C 1× Hanks Balanced Salt Solution (Gibco) supplemented with DMSO or 15 μg/mL CHX (depending on treatment group). We then cultured cells in complete RP10 with DMSO or 15 μg/mL CHX and surveyed cells using flow cytometry at various timepoints. As a positive control, we stimulated CD8$^+$ T cells with 1 μM SIINFEKL peptide (Genemed Synthesis Inc., San Antonio, TX) in complete RP10 in the presence of 1:1000 Golgi plug (BD Biosciences) and DMSO or 15 μg/mL CHX. After 6 h of culture, we harvested cells and conducted flow staining for intracellular cytokines. We employed previously outlined flow cytometric staining protocols for mouse tissues (see ex vivo flow cytometric analysis of mouse tissues) to evaluate CXCR3 expression and cytokine production; flow reagents for CXCR3L and peptide stimulation assays are listed in Supplementary Table 3. Gating strategies are depicted in Supplementary Fig. 11.

**In vitro bystander-activating cytokine stimulations**. We harvested spleen from memory (Fig. 2a) WT and $Cxcr3^{-/-}$ OT-I animals and prepared single-cell suspensions (as earlier described in ex vivo flow cytometric analysis of mouse tissues). After erythrocyte lysis, bulk leukocytes were plated at ~ 1–2 million per well in a 96-well U-bottom tissue culture plate and stimulated with combinations of 100 ng/ml recombinant mouse IL-12p70, IL-15, and IL-18 (BioLegend) in the presence of 1:1000 Golgi plug. Media with 1:1000 Golgi plug, but without cytokines, was used as a negative control. We cultured cells for 6 h at 37 °C 5% $CO_2$ before harvesting for intracellular cytokine staining. We conducted previously stated flow cytometry staining protocols using flow reagents listed in Supplementary Table 4. Gating strategies are depicted in Supplementary Fig. 12.

**IF**. For staining panels that exclude IFNγ, CXCL9, CXCL10, CD11b, or NKG2D, we immediately embedded unfixed tissues on edge in OCT (Sakura Finetek, Torrance, CA). We froze the tissues in the vapor phase of liquid nitrogen before storing the OCT blocks at −80 °C. We cut 8 μm sections of tissues using a Leica CM1950 cryostat (Leica, Wetzlar, Germany), which were dried 18−72 h at RT before being stored at −80 °C. For CD169, CD45.1, and LM Ag staining, we fixed slides in −20 °C acetone for 5 min and allowed slides to dry completely. For CD45.1, Ki-67, and granzyme B staining, we fixed slides in cytofix/cytoperm (BD Biosciences) for 5 min. We washed and rehydrated slides in PBS, blocked endogenous biotin (Vector Labs, Burlingame, CA), and incubated in staining buffer (1× TBS, 5% mouse serum, 5% human serum). All stains were conducted in staining buffer for 1 h at RT or overnight at 4 °C; when staining for intracellular granzyme B and Ki-67, staining buffer was supplemented with 10% perm/wash (BD biosciences) to ensure cell permeabilization. For staining panels that include IFNγ, CXCL9, CXCL10, CD11b, or NKG2D, we fixed tissues overnight in 4 °C cytofix (BD Biosciences) diluted 1:4 in 1× PBS. We then dehydrated fixed tissues in 30% w/v sucrose in 1× PBS for 24 h. We similarly embedded, stored, sectioned, and blocked endogenous biotin in fixed and unfixed tissues. All stains on previously fixed tissues used 1× TBS + 0.3% Triton-X supplemented with 5% mouse serum and 5% human serum as staining buffer. We used 1× TBS + 0.3% Triton-X as a washing buffer. We conducted all stains for 2 h at RT or overnight at 4 °C. Images were collected from an Aperio SlideScan FL or SP8 confocal microscope (Leica Biosystems) using ×20 objectives. IF reagents, dilutions, and fixation/permeabilization methods are listed in Supplementary Tables 5–11.

**Flow cytometric and IF data analysis**. All flow cytometry data was analyzed using FlowJo v9 and v10. Analyzed data were exported for statistical analysis using Prism

v7 and v8 (GraphPad Software, San Diego, CA). Calculations from IF data were all conducted using raw images. Raw IF images acquired on the Aperio SlideScan FL were directly imported into HALO (Indica Labs, Albuquerque, NM) for analysis. Using these, we determined thresholds for positive calls for each staining batch; these values were not changed to accommodate intra-batch variation. We determined WP regions as those circumscribed by CD169 staining (drawn by magnetic gating to CD169) and RP regions as those outside CD169 borders, but within the confines of the spleen (respectively, drawn by magnetic gating to CD169 and DAPI). When CD169 was not included in the staining panel, we imported WP and RP boundaries from a serial section containing CD169. These boundaries were adjusted to WP and RP areas visible by DAPI staining. We defined LM Ag-containing foci as those containing more than five large punctate stains in close proximity. We exported HALO-defined cell counts and phenotypes, as well as area of analyzed regions, into Excel (Microsoft Corporation, Redmond, Washington) to calculate cell densities and phenotype frequencies. We conducted statistical analyses on data using Prism v7 and v8. For serial sections, we modified image orientation in Adobe Photoshop 2018 (Adobe Inc., San Jose, CA) to overlay images separate slides. Contrast was enhanced equally across all images in a display item using Adobe Photoshop 2018 or ImageJ. To increase visibility of punctate stains, we increased pixel size using Adobe Photoshop 2018, or "find edges" command in ImageJ.

**Luminex analysis**. We recovered whole spleens from mock- or WT LM-immunized animals and weighed tissues using an analytical scale. We mechanically dissociated spleens in 500 μL of Luminex tissue buffer (1× PBS supplemented with 0.05% Triton-X and 1% FBS) through a 70 μm nylon filter. To separate debris, we centrifuged samples at $9000 \times g$ for 10 min and aliquoted the supernatant. We stored the supernatants at −80 °C before submitting for analysis. The Immune Monitoring Core from Shared Resources at Fred Hutchinson Cancer Research Center conducted Luminex analysis for IFNγ, GM-CSF, IL-2, IL-6, IL-12p70, IL-15, and TNFα on provided samples.

**Reporting summary**. Further information on research design is available in the Nature Research Reporting Summary linked to this article.

## Data availability
The data that support the findings of this study are available from the corresponding author upon reasonable request. Deidentified HVTN 908/205 data will be made available with approval from HVTN 908/205 study sponsors. Source data are included for Figs. 1j, 2d–f, 3e, 3h–l, 4d, 5d, 6d–g, 6j–m and Supplementary Figures 1c–d, 1g–i, 2c–f, 4a–f, 5b–g, 6a–c, 7a, 7c, 8a–c.

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

## Acknowledgements

We thank Jami R. Erickson (JRE) for technical assistance in tissue harvesting and JRE and Dietmar Zehn for critical review of the manuscript, Jessica L. Swarts and Julia D Berkson for technical assistance in tissue harvesting, Veronica Davé for statistical advice, and Kimberly Smythe and Kimberly Melton for their input with IF assays and HALO analysis. We also thank Dr. Ross Kedl and Cody Rester (University of Colorado, Denver) and Dr. Surojit Sarkar (Seattle Children's Research Institute) for generously providing *Cxcr3*$^{-/-}$ OT-I mice and P14 cells, respectively. We thank the HVTN 205/908 study participants, and thank the HVTN 205/908 Protocol Teams, site staff and the NIAID HVTN for providing clinical specimens and data. N.J.M. is a Leslie and Pete Higgins Achievement Rewards for College Scientists Fellow and Dr. Nancy Herrigel-Babienko Memorial Scholar. This work was supported by NIH grants R01 AI 123323 to M.P. and TL1 TR002318 to N.J.M.

## Author contributions

Conceptualization, N.J.M. and M.P.; methodology, N.J.M. and M.P.; investigation N.J.M.; writing—original draft, N.J.M. and M.P.; writing—review & editing, N.J.M., M.J.M., E.A.N., N.F., M.P.; funding acquisition, M.P.; resources, M.J.M., E.A.N., N.F.; supervision, M.P.

## Competing interests

The authors declare no competing interests.
