## [Peer Review File · Nature Communications]

Reviewers' comments:

Reviewer #1 (Virtual/innate memory, general T cell biology)(Remarks to the Author):

The current study characterizes bystander CD8 cells in the context of infection using primarily mouse TCR transgenic models and assessing local effects via in situ imaging. CXCR3 is demonstrated to play a role in their trafficking and activation. While of potential significance and interest, the study lacks thorough characterization of the bystander phenotype and the flow cytometry differences appear marginal or highly variable making definitive conclusions difficult to draw from. In addition, the characterization of activation versus proliferation is likely very time-dependent and would need more thorough assessment rather than one or two time-points.

Major points:

- 1) The data in figure 1 assessing GZB MFI is showing very marginal effects with extremely small MFI (almost at the level of sensitivity) to make any conclusions from. Showing MFI increases of 20 and making statements of significant differences is a reach given the technical variables of using cryopreserved cells at different time-points and the very few donors assessed. It is also unclear as to how these cells were determined to be bystander (ie non-antigen specific and activated) simply based on GZB. In the absence of antigen-specific response tracking, other markers indicative of bystander activation (notably absence of CD25, PD1, upregulation of NKG2D etc) are needed. It is also unclear given the context of the mouse studies where local expansion and activation is focused on as to how germane these circulating cells are (indeed, it is surprising given the nature and reliance of bystander T cells on the local environment that any are observed in the periphery).
2. The MFI shown regarding the CXCR3 levels is very confusing. The endogenous naive cells clearly show a range and only a percent positive but given the uneven distribution (versus the other populations) it is unclear as to how the MFI values were generated as the flow in C appears log scale and the MFI linear. The huge range of percent positive in the GZB/OT1 populations (ranging from about 80% to about 15%) makes conclusions regarding pattern of expression and whether based on positive versus whole populations impossible. The absence of total numbers of each population to provide context also complicates whether these are truly CXCR3 dim.
3. CXCR3 has been shown to also affect memory generation and this makes interpretation on bystander role difficult. In vitro studies showing bystander effects and baselines on engraftment and memory formation are needed in the adoptive transfer studies.
4. The in situ data shown in Figures 2 and 3 are interesting but need to be backed up with flow cytometry using other markers as mentioned previously and total numbers.
5. The assessment of the proliferative capability of bystander cells is done at 24 hours after infection and negative data obtained. This is likely very contextual and dependent on time. More time points are needed after infection to make a statement on activation versus proliferation. These cells have been shown to expand but clearly also dependent on extent of inflammation and infection.
6. While the supposition is that GZB is allowing for killing capability by the cells, there is no data demonstrating this.
7. The OT1 and GZB co-staining in Figure 3 appears variable as is the ki67/OT1. Use of another marker (CD45.1) would strengthen.

8. The extent of donor cell numbers in Figure 4 is confusing as well after infection. This raises questions on whether the VSV-ova is really cleared despite the time. Given that only 10,000 cells were transferred yet in 4D 35% of the CD8s are of donor, it raises questions on variability (coupled with the 18% CD45.1 shown in 6A) and why so high.

Reviewer #2 (Innate memory, viral immunity)(Remarks to the Author):

In their manuscript, Prlic and colleagues provide evidence that bystander-activated memory CD8 T cells use the chemokine receptor CXCR3 to localize to APCs in the white pulp that have engulfed pathogen. This elegant and concise study incorporates the combination of fluorescence microscopy, iv labeling, and CXCR3-deficient cells to draw their conclusions. There is much unknown about the bystander activation of the immune system during the host-pathogen conflict, and these findings will be of broad interest to those in the field of immunology, in particular T cell and NK cell biologists. I feel that with the edit I've suggested below, this manuscript would be a great fit for publication in Nature Communications.

Specific comments:

Is the increased density/function of memory OT-1 in LM-rich zones dependent on IL-12 (from the infected APC) in addition to CXCR3? The authors can use an IL-12 reporter to examine localization of OT-1 to IL-12 producing cells. Perhaps this would be beyond the scope of the current study, but it would also be interesting to explore this hypothesis using IL-12R-deficient OT-1 cells.

Can the authors include staining for CD69 and CD103 (markers of tissues residency)? Are there Trm among the bystander activated memory OT-1 in the white pulp?

With the endogenous memory CD8 population, is there any way to image their localization within the white pulp? Can the authors look for NKG2D+ CD8+ cells by IF and see if they cluster near the LM-engulfed APC?

With the CXCR3-deficient memory OT-1, is there a defect in protection? The authors can measure CFU within the spleen at early time points after challenge with wildtype LM.

The experiments in Supp Fig 6 are important, and I feel should be included in the main figures (perhaps as Fig 7).

The authors briefly discuss the Teixeira study and how Ag-specificity is required for bystander activation. How do they reconcile this conclusion with their current findings that bystander activation is not antigen specific? Perhaps more discussion is warranted here.

Minor:

In line 250, the authors should clarify that the failed "bystander activation" is referring to failed localization of the T cells to LM-engulfed APCs (since the CXCR3-/- OT-1 are not functionally impaired).

The authors end their manuscript by discussing a competition between antigen-specific naive T cells with bystander-activated memory T cells. They may want to discuss recent findings by Oberle et al and Johnson et al (back to back studies in Cell Reports 2016) in this context – where these groups

observed competition between heterologous naïve and memory T cells of differing Ag-specificities.

Quantification for Fig 6F should be included if possible (similar to figs 2 and 3).

Reviewer #3 (Chemokine, immune trafficking)(Remarks to the Author):

The study by Maurice and colleagues describes the effective use of a low-dose infection model (*Listeria monocytogenes*) to assess local bystander memory T cell activation in the spleen. Activation was measured by the analysis of granzyme B and IFN-gamma expression. The experiments appear to be well controlled and the conclusions sound.

The main conclusion (conferred by the title) is that the CXCR3 is responsible for the recruitment of the memory T cells to the site of infection. It is assumed that the low levels of CXCR3 staining observed on activated memory cells is a direct result of CXCR3 and its ligands having been used to relocate them from the periphery and thus is a measure of CXCR3 having undergone downregulation. This is plausible, given the strict control of CXCR3 replenishment, with data from human T cells suggesting that CXCR3 is not recycled like the majority of chemokine receptors, but sensitivity is restored by de novo synthesis of CXCR3 (Meiser et al, *J. Immunol.* 2008, 180:6713-24). This study should be cited to give more context and the possibility of a similar lack of CXCR3 recycling in mouse T cells should be investigated by flow cytometry following incubation of cells with CXCL9 or CXCL10 coupled with cycloheximide treatment. Reference 33, which is cited to support the authors' hypothesis used human CXCR3 transfectants and did not look at CXCR3 recycling kinetics.

Although a requirement for CXCR3 is inferred by the CXCR3 expression levels and the use of CXCR3 KO mice, it would be prudent to assess the production of CXCL9 and CXCL10 in their infection model to square the circle. The authors cite reference 32 as suggesting that *L. monocytogenes* induces the production of CXCR3 ligands from APCs, but this study used human PBMCs in an in vitro assay, so is not directly comparable to the author's own study. Use of the REX3 reporter mice developed by Andrew Luster would be an elegant solution (Groom et al, *Immunity.* 2012, 37: 1091–1103) although IHC would suffice.

J E Pease

Minor Points

Figure 3 – What I presume to be panel D currently lacks an identifying letter.

Figure 4 – y axis text on some panel E graphs is greyed out and is inconsistent (% of CD8+ and % of CD8s).

Scale bars are missing from Figures 5F and Supp Figures S2B, S3, S6D.

Below please find our point-by-point response to each reviewer – revised sections in the manuscript are highlighted **in bright yellow** in the text:

Reviewer 1

Major points:

1) The data in figure 1 assessing GZB MFI is showing very marginal effects with extremely small MFI (almost at the level of sensitivity) to make any conclusions from. Showing MFI increases of 20 and making statements of significant differences is a reach given the technical variables of using cryopreserved cells at different time-points and the very few donors assessed. It is also unclear as to how these cells were determined to be bystander (ie non-antigen specific and activated) simply based on GZB. In the absence of antigen-specific response tracking, other markers indicative of bystander activation (notably absence of CD25, PD1, upregulation of NKG2D etc) are needed. It is also unclear given the context of the mouse studies where local expansion and activation is focused on as to how germane these circulating cells are (indeed, it is surprising given the nature and reliance of bystander T cells on the local environment that any are observed in the periphery).

We agree with the reviewer that a careful analysis and proper controls are important when interpreting human data that are inherently noisier than data from inbred SPF mice. First, we want to highlight that we used the statistical median when assessing fluorescent intensity. Given that granzyme expression is

not normally distributed, using the median is the correct statistical approach, but it also ensures that that changes are not simply driven by a few outlying data points. We show in Fig. 1B that several donors had nearly a doubling in the median fluorescent intensity of granzyme B expression in their memory CD8 T cells, but not in the naïve T cell compartment (Suppl Fig 1B). Importantly, the study design of this vaccine trial also allowed us to include a “placebo” control group and we did not observe granzyme MFI changes in the CD8 memory compartment of the placebo group (Fig. 1B). Together, these data allowed us to conclude that the increase in granzyme B expression in vaccine cohort is specific to the memory compartment.

We also appreciate the suggestions for biomarkers to determine if T cells were recently activated in a TCR-dependent manner. We did not use NKG2D since it is expressed on all human T cells, but now also show PD-1 (as requested by the reviewer) and 4-1BB expression data in the revised manuscript (Suppl. Fig. 1C-E). Briefly, Greenberg and colleagues demonstrated that 4-1BB is expressed following TCR engagement on memory T cells (Ref. 21. Wöfl, *et al. Cytometry A* 2008 DOI: 10.1002/cyto.a.20594). In line with our previous conclusions we did not observe an increase in PD-1 or 4-1BB expression. Together these data suggest that granzyme B expression is indeed driven by inflammatory cues and not TCR-mediated signals.

Finally, we show in our manuscript that the initial bystander activation event occurs in a localized manner, but it is important to keep in mind that these bystander-activated cells can leave this initial site of activation.

2. The MFI shown regarding the CXCR3 levels is very confusing. The endogenous naïve cells clearly show a range and only a percent positive but given the uneven distribution (versus the other populations) it is unclear as to how the MFI values were generated as the flow in C appears log scale and the MFI linear. The huge range of percent positive in the GZB/OT1 populations (ranging from about 80% to about 15%) makes conclusions regarding pattern of expression and whether based on positive versus whole populations impossible. The absence of total numbers of each population to provide context also complicates whether these are truly CXCR3 dim.

We changed the scale of our graphs from linear to log as requested by the reviewer. As pointed out by this reviewer there is an uneven distribution in CXCR3 expression, which is why we used MFI values that are based on the median (and not mean). The CXCR3 median fluorescent intensity of each T cell population of each animal is shown in the left panel in Fig. 5D and % CXCR3⁺ cells are shown in the right panel. Both, changes in MFI and % of CXCR3 expression are very pronounced and also highly statistically significant. Finally, we performed additional experiments to directly demonstrate that exposure to CXCL9 and CXCL10 decreased CXCR3 expression in a dose-dependent manner (new Fig. 6C, D) and we enumerated the CXCR3dim cells as requested (Suppl. Fig. 6C)

3. CXCR3 has been shown to also affect memory generation and this makes interpretation on bystander role difficult. In vitro studies showing bystander effects and baselines on engraftment and memory formation are needed in the adoptive transfer studies.

We performed additional experiments to test the ability of CXCR3^{-/-} memory T cells to respond to bystander activation in vitro (Fig. 7J-M, Suppl. Fig. 8B, C). Our data indicate that CXCR3^{-/-} memory T cells respond to pro-inflammatory signals and become bystander-activated in vitro demonstrating that they are not inherently defective (in line with our initial conclusions and Fig. 7G). The requested WT vs. CXCR3^{-/-} memory formation data are now shown in Suppl. Fig. 8A. These data are in concordance with published literature cited in our manuscript (Ref. 37, Kurachi, *et al. J Exp Med.* 2011 DOI: 10.1084/jem.20102101), demonstrating that CXCR3-deficient T cells form a larger memory population, yet remain functionally viable.

4. The in situ data shown in Figures 2 and 3 are interesting but need to be backed up with flow cytometry using other markers as mentioned previously and total numbers.

We thank the reviewer for this comment and included additional data in Suppl. Fig. 4D.

5. The assessment of the proliferative capability of bystander cells is done at 24 hours after infection and negative data obtained. This is likely very contextual and dependent on time. More time points are

needed after infection to make a statement on activation versus proliferation. These cells have been shown to expand but clearly also dependent on extent of inflammation and infection.

We agree with the reviewer that the onset of proliferation is highly time dependent. We specifically asked if the increase of bystander-activated T cells around foci of infection at the 24hr time-point is due to proliferation or migration. Our data showed that this early, 24hr clustering is due to migration (Fig. 3). As requested by the reviewer, we added additional time-points and found that cells become Ki67⁺ at later time points as expected (based on the initial bystander papers published by Tough & Sprent, Ref. 1 in our manuscript). These data are shown in Suppl Fig. 4C and D.

6. While the supposition is that GZB is allowing for killing capability by the cells, there is no data demonstrating this.

We apologize that we did not make it more clear that we have already previously demonstrated that bystander-activated T cells can kill in an NKG2D-dependent manner (Ref. 7, Chu et al., *Cell Reports* 2013, DOI: 10.1016/j.celrep.2013.02.020). We revised the manuscript accordingly.

7. The OT1 and GZB co-staining in Figure 3 appears variable as is the ki67/OT1. Use of another marker (CD45.1) would strengthen.

We apologize, but it is unclear to us what the specific question is. We used CD45.1 in all of our IF experiments to identify OT-I T cells. IF and flow cytometry data for Ki67 and granzyme B are consistent. Additional time-points are now shown in Suppl. Fig. 4C and D.

8. The extent of donor cell numbers in Figure 4 is confusing as well after infection. This raises questions on whether the VSV-ova is really cleared despite the time. Given that only 10,000 cells were transferred yet in 4D 35% of the CD8s are of donor, it raises questions on variability (coupled with the 18% CD45.1 shown in 6A) and why so high.

We chose VSV-OVA, because it is a well-established system to generate OT-I memory T cells. Recombinant VSV is quickly eliminated in mice, which has been shown by John Rose's group (Simon et al., *J. Virol*, 2010 DOI: 10.1128/JVI.02052-09) and others. We and others (numerous reports from the Lefrancois and Bevan labs) have previously shown that adoptively transferred naïve OT-I T cells expand and form a stable and sizeable memory population following infection with VSV-OVA. Suppl. Fig. 8A provides an overview of this OT-I memory population and is well in line with these previous studies in regards to initial expansion and subsequent maintenance.

Reviewer #2

Specific comments:

Is the increased density/function of memory OT-1 in LM-rich zones dependent on IL-12 (from the infected APC) in addition to CXCR3? The authors can use an IL-12 reporter to examine localization of OT-1 to IL-12 producing cells. Perhaps this would be beyond the scope of the current study, but it would also be interesting to explore this hypothesis using IL-12R-deficient OT-1 cells.

We agree with the reviewer that this is an interesting question. We approached it by first asking if IL-12 is necessary or sufficient for bystander activation. We found that IL-12 was not necessary but enhanced the effect of IL-15 and IL-18 on memory T cells (new Fig. 7J-M, Suppl. Fig. 8B, C). Given that IL-12 was not necessary we did not pursue the IL-12R^{-/-} experiment. Importantly, since IL-12 enhanced the effect of IL-15 and IL-18 *in vitro*, we asked if the *in vivo* IL-12 production correlated with IFN γ production. We tested this by examining spleens by Luminex 24 hours post-infection with LM and found a near-perfect correlation (new Fig. 4 K-L).

Can the authors include staining for CD69 and CD103 (markers of tissues residency)? Are there Trm among the bystander activated memory OT-1 in the white pulp?

We thank the reviewer for the interesting suggestion. We performed several experiments to address the question. Interestingly, we found that the Trm phenotype cells had a much lower response to bystander-mediated activation compared to their non-Trm counterparts (shown in the new Suppl. Fig. 5D-G). We could not use CD69 alone as a biomarker since both inflammatory and TCR signals can induce CD69

expression (Suppl. Fig. 5A, B; the observation that inflammatory signals can induce CD69 expression was initially made by Hao Shen and reported in *J. Immunol.*, DOI: [10.4049/jimmunol.171.8.4352](https://doi.org/10.4049/jimmunol.171.8.4352)), but we could use CD69 together with CD103 as a marker for Trm. While outside the scope of this study, we think that it will be worthwhile to follow up on this observation and examine Trm cells in other tissues to determine if Trm are less responsive to bystander-activation regardless of their tissue-origin.

With the endogenous memory CD8 population, is there any way to image their localization within the white pulp? Can the authors look for NKG2D+ CD8+ cells by IF and see if they cluster near the LM-engulfed APC?

We thank the reviewer for the suggestion. We attempted the experiment as described by the reviewer, however CD8 staining failed to work in slides fixed for NKG2D staining. We utilized NKG2D, granzyme B, and CD45.1 (for OT-I) staining (new Fig. 4N). Here we found that a majority of clustering NKG2D+ CD45.1- cells express granzyme B. Despite the presence of NK cells which, too, stain for NKG2D, we believe these data complement our flow cytometry data in Fig. 4E, H-J. We similarly stained sections for IFN γ , granzyme B, and CD45.1 (new Fig. 4M). Here we found both OT-I and CD45.1⁺ cells within clusters producing IFN γ , further supporting our interpretation of the flow data.

With the CXCR3-deficient memory OT-1, is there a defect in protection? The authors can measure CFU within the spleen at early time points after challenge with wildtype LM.

This is an interesting question that is unfortunately complicated by the presence of endogenous memory T cells in the adoptive transfer set-up. Specifically, endogenous virtual memory T cells are potent at becoming bystander-activated (demonstrated by Kedl and colleagues, White et al., *Nat Commun* 2016) and these endogenous virtual memory T cells are substantial in number. We considered an experiment with CXCR3^{-/-} mice, but we hope that the reviewer agrees with our conclusion that the data from the these knock-out mice would be uninterpretable given that other immune subsets (incl. myeloid cells) also express CXCR3.

The experiments in Supp Fig 6 are important, and I feel should be included in the main figures (perhaps as Fig 7).

We thank the reviewer for this comment and now include the data from previous Supp Fig. 6 in the main figures (new Fig. 7G)

The authors briefly discuss the Teixeira study and how Ag-specificity is required for bystander activation. How do they reconcile this conclusion with their current findings that bystander activation is not antigen specific? Perhaps more discussion is warranted here.

We apologize for the confusion. Teixeira and colleagues showed that while antigen is not required, the presence of self-peptide and MHC class I was needed. We think that this observation is really interesting as it suggests that NK cells and bystander-activated T cells recognize different NKG2D ligand-expression target cells depending on the presence or absence of MHC class I.

Minor:

In line 250, the authors should clarify that the failed “bystander activation” is referring to failed localization of the T cells to LM-engulfed APCs (since the CXCR3^{-/-} OT-1 are not functionally impaired).

This is a really important point and we edited the sentence as suggested by the reviewer.

The authors end their manuscript by discussing a competition between antigen-specific naïve T cells with bystander-activated memory T cells. They may want to discuss recent findings by Oberle et al and Johnson et al (back to back studies in *Cell Reports* 2016) in this context – where these groups observed competition between heterologous naïve and memory T cells of differing Ag-specificities.

We thank the reviewer for the helpful suggestion. We revised the manuscript to briefly summarize these papers and include a discussion about the importance of antigen availability for primary T cell responses with an emphasis on how bystander-activated T cells may diminish antigen availability.

Quantification for Fig 6F should be included if possible (similar to figs 2 and 3).

We thank the reviewer for the comment. We now include flow cytometry based quantification to complement the IF data.

Reviewer #3

[...] CXCR3 is not recycled like the majority of chemokine receptors, but sensitivity is restored by *de novo* synthesis of CXCR3 (Meiser et al, J. Immunol. 2008, 180:6713-24). This study should be cited to give more context and the possibility of a similar lack of CXCR3 recycling in mouse T cells should be investigated by flow cytometry following incubation of cells with CXCL9 or CXCL10 coupled with cycloheximide treatment. Reference 33, which is cited to support the authors' hypothesis used human CXCR3 transfectants and did not look at CXCR3 recycling kinetics.

We agree with the reviewer and performed the suggested experiment (incubate memory T cells with CXCL9/CXCL10 with or without cycloheximide). These data are now included as new Fig. 6C, D and Suppl. Fig. 7A-C). Briefly, we titrated CXCL9/CXCL10 and found that CXCR3 expression on the surface decreases in a dose-dependent manner within 30min (Fig. 6C, D; Suppl. Fig. 7A). Importantly, when the ligands are removed by washing, CXCR3 expression returns to baseline within 90min even in the presence of cycloheximide. These data suggest that following internalization mouse CXCR3 is recycled back to the surface and does not depend on *de novo* protein synthesis. We ensured that the cycloheximide worked by included a proper positive control (Suppl. Fig. 7E)

Although a requirement for CXCR3 is inferred by the CXCR3 expression levels and the use of CXCR3 KO mice, it would be prudent to assess the production of CXCL9 and CXCL10 in their infection model to square the circle. The authors cite reference 32 as suggesting that *L. monocytogenes* induces the production of CXCR3 ligands from APCs, but this study used human PBMCs in an in vitro assay, so is not directly comparable to the author's own study. Use of the REX3 reporter mice developed by Andrew Luster would be an elegant solution (Groom et al, Immunity. 2012, 37: 1091–1103) although IHC would suffice.

We agree and thank the reviewer for the suggestions. We changed the reference and performed several new experiments to detect CXCL9 and CXCL10 by immunofluorescence (IF). We chose IF, because we were concerned that requesting the reporter mice, getting them through institutional quarantine, then setting up breeders and waiting for the litters to be at least 6 weeks old would not be feasible for the requested 3-month timeframe to return revisions. We are happy to report that the IF approach worked really well once optimized (the protocol is included in the material and methods). We found that CXCL9 and CXCL10 are produced around *Listeria*-infected cells in the white pulp (see new Fig. 6E), which is in line with our previous conclusions.

Minor Points

Figure 3 – What I presume to be panel D currently lacks an identifying letter.

We apologize for the omission – this error is now fixed.

Figure 4 – y axis text on some panel E graphs is greyed out and is inconsistent (% of CD8+ and % of CD8s). Scale bars are missing from Figures 5F and Supp Figures S2B, S3, S6D.

We apologize for the inconsistencies – these are all fixed in the revised manuscript.

Reviewers' comments:

Reviewer #1 (Remarks to the Author):

The revised study is somewhat improved by there are still significant issues with the human data in figure 1. It is clear looking at the axis that the data shown in 1B are MFI of the entire cell population of which over 70-90% of the gated CD8+ T cells are negative for GZB judging from 1C. There are orders of magnitude differences between the MFI in 1B versus 1C. It is therefore critical, given that the data in 1B are the only significantly different values to indicate why not addressing MFI in the positive population was not shown. The shift in MFI in 1B since incredibly small is likely based on what would be considered "negative" populations and misleading given the primary criteria for bystander in the study GZB positivity.

The other issue with the supplemental data is that a very different vaccination regimen (the schema is very confusing; is everyone a "placebo"?) is used so it is not clear if the data in Figure 1 is actually pertinent to the data in the supplemental. Furthermore, the baseline PD1 staining in either cohort appears extremely high (60% of the RO+) and uniformly so.

Finally, the mouse data do not link at all with the human as GZB MFI never shown but % differences instead and infection models are very different than vaccine ones. In that regard, the data are tenuous given the very small sample size and source of cells (ie PBMC as cells in transit versus local as in the mouse models).

The mouse data are solid and provide interesting new paradigm in bystander immunobiology.

Reviewer #2 (Remarks to the Author):

The authors have done a nice job of addressing all of my concerns with either new experiments or further discussion. This is a solid study that addresses an outstanding question in the field. There is much unknown about the bystander activation of the immune system during host-pathogen conflict, and these findings will be of broad interest to those in the field of immunology. I feel this manuscript is ready for publication in Nature Communications.

Reviewer #3 (Remarks to the Author):

I am happy that the authors have responded positively to my comments and have carried out additional experimental work which I think adds to their story.

We are happy that reviewer 1 stated that the “mouse data are solid and provide an interesting new paradigm” in line with the conclusions from reviewers 2 and 3 (“a solid study”, “these findings will be of broad interest to those in the field of immunology”). We are pleased that we could address all comments provided by reviewers 2 and 3 to their satisfaction and we thank all three reviewers for their constructive and helpful comments.

Below please find our point-by-point response to the last remaining comments are all in regarding to the human data included in our manuscript (Fig. 1 and Suppl. Fig. 1) – revised sections in the manuscript are highlighted **in bright yellow** in the text:

Briefly, the following statement provided by reviewer 1 was eye-opening for us “The other issue with the supplemental data is that a very different vaccination regimen (the schema is very confusing; is everyone a "placebo"?) is used so it is not clear if the data in Figure 1 is actually pertinent to the data in the supplemental.”, because it explained previous and current questions provided by reviewer 1. Importantly, we rewrote a large part of the Figure 1 results section and edited Supplemental Figure 1 to make it clear that the vaccine group and the placebo group are from the same clinical trial (HVTN908). We hope that reviewer 1 agrees that this revised version now makes it clear that the placebo (saline) group is a suitable control for the vaccine (MVA) group as well as for our analysis. We apologize for any confusion that our initial vaccination scheme in Suppl Fig 1 may have caused.

The revised study is somewhat improved by there are still significant issues with the human data in figure 1. It is clear looking at the axis that the data shown in 1B are MFI of the entire cell population of which over 70-90% of the gated CD8+ T cells are negative for GZB judging from 1C. There are orders of magnitude differences between the MFI in 1B versus 1C. It is therefore critical, given that the data in 1B are the only significantly different values to indicate why not addressing MFI in the positive population was not shown. The shift in MFI in 1B since incredibly small is likely based on what would be considered "negative" populations and misleading given the primary criteria for bystander in the study GZB positivity. We hope that the revisions mentioned above address most of the comments provided by reviewer 1. Importantly, we used the entire CD8 memory population to calculate the median fluorescence intensity (MFI) to avoid introducing a potential bias by arbitrarily setting a “positive” gate. Furthermore, we chose to show the median (not mean!) change of granzyme B expression in the entire memory CD8 T cell population (in vaccine vs. placebo), because we have no a priori knowledge of which memory CD8 T cells increase granzyme B expression. We considered that some memory CD8 T cells could increase granzyme expression from low to medium levels, which we can only assess by measuring MFI changes in the entire population. Since the median expression levels of granzyme B change

exclusively in the vaccine but not the placebo cohort and considering that we are measuring this effect across the entire memory CD8 T cell compartment (pre-immunization/day 0 vs. day 3 post-immunization), we argue that the increase in granzyme B expression following immunization is not only statistically but also biologically significant. With that said, we edited our manuscript to tone down our conclusions (see highlighted text in the results and discussion section) and hope this will now fully address the reviewer's comment.

Furthermore, the baseline PD1 staining in either cohort appears extremely high (60% of the RO+) and uniformly so.

PD-1 expression by CD8 T cells is certainly cohort-dependent, but our data are very similar in that regard to a published study from Rafi Ahmed and colleagues (J. Immunol, 2011, DOI: <https://doi.org/10.4049/jimmunol.1001783>) that also reported that about 60% of the memory CD8+ T cells express PD-1. Importantly, they go on to show that PD-1 expressing cells are effector memory cells (but NOT exhausted or dysfunctional).

Finally, the mouse data do not link at all with the human as GZB MFI never shown but % differences instead and infection models are very different than vaccine ones. In that regard, the data are tenuous given the very small sample size and source of cells (ie PBMC as cells in transit versus local as in the mouse models).

The main point of the human data is to demonstrate that bystander-activation can be observed even after providing a fairly localized immune stimulus (MVA-based vaccine, i.m.). This is noteworthy, because in the past bystander-activation was always associated with big, systemic immune insults.

We edited the manuscript to make it more clear that we include the human data to provide relevance (i.e. this is not just a mouse model phenomenon). We are not trying to directly compare the response kinetics between an i.v. immunization mouse model and a human i.m. -administered vaccine. We hope that our revisions will fully address this comment.

REVIEWERS' COMMENTS:

Reviewer #1 (Remarks to the Author):

The revised manuscript is improved but I must take issue with the arguments regarding the MFI data in the first figure. The median is indeed appropriate and that the MFI of the total population can eliminate the subjectivity of gating differences but it simply has to be put in context of positivity and negativity based on appropriate internal (FMO, isotype) and external (sampling and handling techniques) controls. To say that cells in the negative gating population are "low to medium" expressors as reflected by antibody binding obviates the use of specific antibodies to precisely address whether expression is occurring. Instead, degrees of expression are based on such gating (ie high versus low expression is still based on detection above background) via control/background levels to demonstrate presence of the marker in question. Stressing the importance of a shift in MFI between samples when there is no difference in positivity (or MFI of the positively staining cells) does not make sense as pertains to degrees of negativity. As GZB is proposed to be a marker for this memory T cell population being bystander activated it is therefore important to examine the GZB+ memory T cell population not all memory as that is not what is being assessed. It simply does not make sense that even the negative staining population based on gating is therefore also to be considered positive yet also a marker for bystander activated T cells.